# Altered TMPRSS2 usage by SARS-CoV-2 Omicron impacts infectivity and fusogenicity

Bo Meng[1,2,50], Adam Abdullahi[1,2,50], Isabella A. T. M. Ferreira[1,2,50], Niluka Goonawardane[1,2,50], Akatsuki Saito[3,50], Izumi Kimura[4,50], Daichi Yamasoba[4,50], Pehuén Pereyra Gerber[1,2], Saman Fatihi[5], Surabhi Rathore[5], Samantha K. Zepeda[6], Guido Papa[7], Steven A. Kemp[1,2], Terumasa Ikeda[8], Mako Toyoda[9], Toong Seng Tan[9], Jin Kuramochi[10], Shigeki Mitsunaga[11], Takamasa Ueno[9], Kotaro Shirakawa[12], Akifumi Takaori-Kondo[12], Teresa Brevini[2], Donna L. Mallery[7], Oscar J. Charles[13], The CITIID-NIHR BioResource COVID-19 Collaboration*, The Genotype to Phenotype Japan (G2P-Japan) Consortium*, Ecuador-COVID19 Consortium*, John E. Bowen[6], Anshu Joshi[6], Alexandra C. Walls[6,14], Laurelle Jackson[15], Darren Martin[16], Kenneth G. C. Smith[1,2], John Bradley[2], John A. G. Briggs[17], Jinwook Choi[18], Elo Madissoon[19,20], Kerstin B. Meyer[19], Petra Mlcochova[1,2], Lourdes Ceron-Gutierrez[21], Rainer Doffinger[21], Sarah A. Teichmann[19,22], Andrew J. Fisher[23], Matteo S. Pizzuto[24], Anna de Marco[24], Davide Corti[24], Myra Hosmillo[14], Joo Hyeon Lee[18,25], Leo C. James[7], Lipi Thukral[5], David Veesler[6,26], Alex Sigal[15,27,28], Fotios Sampaziotis[1,2,18,21], Ian G. Goodfellow[14], Nicholas J. Matheson[1,2,21,29], Kei Sato[4,30✉] & Ravindra K. Gupta[1,2,15✉]

The SARS-CoV-2 Omicron BA.1 variant emerged in 2021[1] and has multiple mutations in its spike protein[2]. Here we show that the spike protein of Omicron has a higher affinity for ACE2 compared with Delta, and a marked change in its antigenicity increases Omicron's evasion of therapeutic monoclonal and vaccine-elicited polyclonal neutralizing antibodies after two doses. mRNA vaccination as a third vaccine dose rescues and broadens neutralization. Importantly, the antiviral drugs remdesivir and molnupiravir retain efficacy against Omicron BA.1. Replication was similar for Omicron and Delta virus isolates in human nasal epithelial cultures. However, in lung cells and gut cells, Omicron demonstrated lower replication. Omicron spike protein was less efficiently cleaved compared with Delta. The differences in replication were mapped to the entry efficiency of the virus on the basis of spike-pseudotyped virus assays. The defect in entry of Omicron pseudotyped virus to specific cell types effectively correlated with higher cellular RNA expression of *TMPRSS2*, and deletion of *TMPRSS2* affected Delta entry to a greater extent than Omicron. Furthermore, drug inhibitors targeting specific entry pathways[3] demonstrated that the Omicron spike inefficiently uses the cellular protease TMPRSS2, which promotes cell entry through plasma membrane fusion, with greater dependency on cell entry through the endocytic pathway. Consistent with suboptimal S1/S2 cleavage and inability to use TMPRSS2, syncytium formation by the Omicron spike was substantially impaired compared with the Delta spike. The less efficient spike cleavage of Omicron at S1/S2 is associated with a shift in cellular tropism away from TMPRSS2-expressing cells, with implications for altered pathogenesis.

The Omicron variant of SARS-CoV-2, which was first detected in South Africa[1], carries over 30 mutations in its spike protein and has now spread internationally at a fast pace. More than 20 substitutions exist in the N-terminal domain and receptor-binding domain (RBD). The Omicron variant also contains six unique mutations in S2 (N764K, D796Y, N856K, Q954H, N969K and L981F) that were not previously detected in other variants of concern (VOCs). The spike proteins of the Alpha and Delta variants were previously shown to confer more efficient cell–cell fusion kinetics compared with Wuhan-Hu-1 (refs. [4,5])

due to mutations in the furin-cleavage site region that increase S1/S2 cleavage. Furthermore, Alpha and Delta showed more efficient syncytia formation, which was previously found to be associated with pathogenesis[6]. Omicron has three mutations in the region of the furin-cleavage site (P681H, H655Y and N679K) and was initially predicted to be highly infectious[7] and pathogenic. Indeed, Omicron has been associated with very rapid increases in case numbers and recent data demonstrate considerable reinfection and vaccine 'breakthrough', probably due to evasion of neutralizing antibody

---

A list of affiliations appears at the end of the paper. *A list of authors and their affiliations appears at the end of the paper.

responses[2,8]. However, paradoxically, recent findings suggest reduced severity in Omicron infections compared to Delta[9].

Here we examine the biological properties of Omicron, and focus on spike-mediated evasion of neutralizing antibodies, increased ACE2 binding affinity, a shift in tropism away from TMPRSS2-expressing cells and compromised ability to generate syncytia.

## Omicron spike-binding affinity to ACE2

The Omicron variant has 15 amino acid mutations in the RBD (Supplementary Figs. 1 and 2). To understand the impact of these substitutions on receptor engagement, we determined the kinetics and affinity of monomeric human ACE2 binding to immobilized Omicron, Wuhan-Hu-1 and Delta RBDs using biolayer interferometry. We observed that the Omicron RBD has approximately threefold enhanced binding affinity for ACE2 relative to the Wuhan-Hu-1 and Delta RBDs (Extended Data Table 1 and Supplementary Fig. 2), consistent with previous findings[10–13]. As previously observed for the Alpha RBD, which contains only the N501Y mutation[14], the modulation of binding is mediated by changes in ACE2 dissociation rates. Although the K417N mutation is known to decrease ACE2 engagement[14,15], a recently determined crystal structure of ACE2-bound Omicron RBD revealed that Q493R and Q498R introduce additional electrostatic interactions with ACE2 residues Glu35 and Asp38, respectively[15], whereas S477N enables hydrogen-bonding with ACE2 Ser19 (Supplementary Fig. 2). Collectively, these mutations strengthen ACE2 binding, relative to the ancestral isolate. We extended this binding analysis to a cell-based model using cells that were transfected with full-length spike, followed by ACE2 antibody titration; we observed significantly higher ACE2 binding for Omicron spike compared with the spike proteins of the Wuhan-Hu-1 and Delta variants (Supplementary Fig. 2). This enhanced binding could be a factor in the enhanced transmissibility of Omicron relative to previous variants.

## The sensitivity of Omicron to therapeutics

Omicron was predicted to have broad resistance to neutralizing antibodies based on mutations in the class I–IV antigenic regions in its RBD[16]. Current standard of care antiviral treatment for moderate to severe COVID-19 includes the use of the monoclonal antibody combination REGN10933 (casivirimab) and REGN10897 (imdevimab). The K417N, E484A, S477N and Q493R mutations preclude REGN10933 binding, whereas G446S clashes with REGN10987, consistent with the decreased binding to the Omicron RBD and S trimer and the loss of neutralization previously descibed[11,15]. We next tested serial dilutions of component monoclonal antibodies, both individually and in combination, against Delta and Omicron live viruses in tissue culture (Fig. 1a). Although the Delta variant was effectively neutralized by casirivimab, imdevimab was only partially effective, consistent with previous data[4]. In combination, these monoclonal antibodies were highly potent against Delta. However, there was a complete loss of neutralizing activity against Omicron by these monoclonal antibodies alone or in combination (Fig. 1a). Given these results, we next tested the direct-acting antivirals remdesivir and the active metabolite of molnupiravir against live viruses. We observed similar antiviral activity against Delta and Omicron using serial titrations of both compounds (Supplementary Fig. 3).

## Omicron and polyclonal antibodies

An important question is whether vaccine-elicited antibodies are able to neutralize Omicron. We synthesized codon-optimized spike expression plasmids for Omicron and Delta spike proteins and generated pseudotyped virus (PV) particles by co-transfecting the spike expression plasmids with a lentiviral gag-pol-expressing plasmid and a lentiviral transduction plasmid encoding the luciferase gene[17,18]. We obtained longitudinal serum samples from 40 individuals who were vaccinated with either the BNT162b2 or ChAdOx-1 vaccine and performed serum titrations before mixing sera with our reporter PV particles. The participants had a median age of around 70 years and prospective serum samples were taken as follows: one month after dose two, six months after dose two and one month after dose three (Extended Data Table 2). We observed at least tenfold loss of neutralization against Omicron after the second dose compared with Delta (Fig. 1b, c and Extended Data Fig. 4). Indeed, neutralization of Omicron was not detectable for the majority of individuals who had received two doses of ChAdOx-1. We also observed waning over time after the second dose for both vaccines (Fig. 1b, c). Both groups received a booster dose of BNT162b2 as a third dose, enabling us to compare the response to this booster dose. Considerable increases (greater than tenfold) in neutralization against both Omicron and Delta were observed for all of the variants after the third dose of vaccination, suggesting an increase in the breadth of responses as well as titre.

To confirm the loss of neutralizing activity against Omicron after the second dose, we next used a live virus experimental system to compare Delta and Omicron variants against sera taken four weeks after the second dose of BNT162b2, and obtained similar results to those obtained in the PV assay (Extended Data Fig. 5a). These live viruses were also used to assess the neutralization of the Omicron variant by sera derived from unvaccinated individuals who were previously infected with the early Wuhan-Hu-1 virus or the Delta variant. As expected, vaccine sera had significantly impaired activity against Omicron compared with Delta (Extended Data Fig. 5a). We also tested mRNA-1273 vaccine-elicited sera, which showed a similar reduction in neutralization to BNT162b2. However, coronavac sera showed little neutralization against Delta and 0 out of 9 participants had detectable neutralization against Omicron. Sera from Delta infections appeared to have lower cross-neutralization compared with those from the early pandemic period when Wuhan-Hu-1 D614G was dominant (Extended Data Fig. 5b).

## Omicron replication and spike cleavage

We infected primary human nasal epithelial 3D cultures (hNECs) with an air–liquid interface (Fig. 2a, b). Infection with live SARS-CoV-2 Omicron or Delta was conducted at the apical surface with equal amounts of input virus and virus collected from the apical surface at 24 and 48 h, and virus was quantified using quantitative PCR (qPCR) analysis of the $E$ gene and 50% tissue culture infectious dose (TCID$_{50}$). We observed similar replication kinetics for Omicron and Delta in the hNECs (Fig. 2b). We next infected Calu-3 lung cells (which are known to express endogenous *TMPRSS2*) with live isolates, and observed significantly greater viral replication for Delta than Omicron (Fig. 2c), manifesting as early as 24 h after infection. We also infected Caco-2 cells, a colon cancer cell line (known to express high levels of endogenous *TMPRSS2*)[19] and, as for Calu-3, we found 1 log greater Delta RNA and higher infectious virus (TCID$_{50}$) in the supernatants compared with those observed for Omicron. Finally, we tested a cell line—HeLa, overexpressing *ACE2* and *TMPRSS2*—and obtained similar results to those of the Calu-3 and Caco-2 cells.

We next sought to examine the possibility that the impaired replication in some cells may be related to differential spike incorporation and/or cleavage. For example, Delta, which is known to have higher replication, is associated with a highly cleaved spike protein[4]. We tested purified virions from two independent Omicron isolate infections. We observed broadly similar amounts of Omicron and Delta spike incorporation into virions generated in VeroE6 ACE2/TMPRSS2 cells (Fig. 2f). However, there was reduced cleavage in whole Omicron virions compared with Delta as evidenced by the ratio of full-length spike to S1 and S2; the Omicron spike was inefficiently cleaved in virions compared with

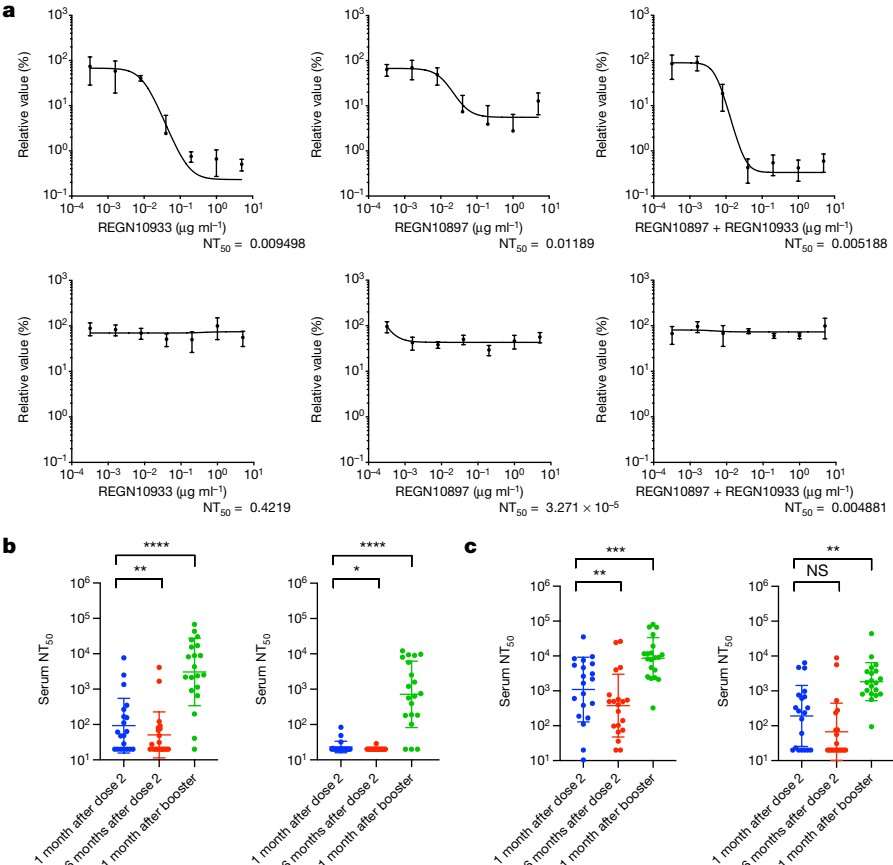

**Fig. 1 | The sensitivity of SARS-CoV-2 Omicron to clinically approved monoclonal antibodies and to vaccine-elicited neutralizing antibodies.** **a**, Titration of the monoclonal antibodies REGN10933 and REGN10897 individually and a combination of both against replication-competent Delta (top) and Omicron (bottom) viruses. Data are mean ± s.e.m. $n = 4$ of technical replicates at each dilution for each virus. Curve fitting for dose response was performed in GraphPad Prism. $NT_{50}$, 50% neutralization titre. **b**, **c**, Neutralization of Delta (left) and Omicron (right) spike PV by sera from vaccinated individuals (ChAdOx-1 (**b**; $n = 20$) or BNT12b2 (**c**; $n = 20$)) over three time points after dose two (ChAdOx-1 or BNT162b2) and dose three (BNT162b2 only). Data are the geometric mean titre ± s.d., representative of two independent experiments each with two technical replicates. Statistical analysis was performed using Wilcoxon matched-pairs signed-rank tests; \*\*$P < 0.01$, \*\*\*$P < 0.001$, \*\*\*\*$P < 0.0001$; NS, not significant.

Delta (Fig. 2f). However, more substantially reduced Omicron spike cleavage compared with Delta was observed in cell lysates (Fig. 2g).

Given the lower intracellular S1/S2 cleavage for Omicron versus Delta spike proteins, confocal microscopy was used to examine whether cleavage efficiency is related to differences in subcellular localization. We analysed the distribution of Spike in cells infected with Omicron and Delta SARS-CoV-2 live virus isolates (Fig. 2h–j). No clear differences were observed between the variants in distance from the nucleus or co-localization with a Golgi marker, indicating that the subcellular localization between Delta and Omicron is similar.

## Omicron entry is impaired in TMPRSS2⁺ cells

SARS-CoV-2 spike is a major determinant of viral infectivity and mediates cell entry through interaction with ACE2 (refs. [3,10]) and either TMPRSS2 (ref. [3]) at the plasma membrane or cathepsins in the endosomes. The plasma membrane route of entry, and indeed transmissibility and disease severity in animal models, is critically dependent on the polybasic cleavage site (PBCS) between S1 and S2 (refs. [6,7]) and cleavage of spike before virion release from producer cells; this contrasts with the endosomal entry route, which does not require spike cleavage in producer cells[3]. Plasma membrane fusion also enables the virus to avoid restriction factors in the endosomes[7].

We tested spike-mediated viral entry of the wild-type (WT) Wuhan-Hu-1 D614G, Delta and Omicron spike proteins (Fig. 3a) using

the PV system. We first probed PV virions for spike protein and, after western blot analysis, noted a reduced incorporation of Omicron spike into virions compared with the incorporation of Delta spike (Extended Data Fig. 6). We also noted that the Omicron spike was less cleaved than the Delta spike, as observed with the live virus (Extended Data Fig. 6). Notably, cleavage of Omicron spike in cells was also lower compared with Delta and WT spike, again similar to that in the live virus.

Using normalized amounts of Delta and Omicron spike PV, we infected primary 3D lower-airway organoids and gallbladder organoids[20,21] (Fig. 3a and Extended Data Fig. 6), in addition to Calu-3 lung cells. Mirroring the lower replication of Omicron relative to Delta in live virus assays, we observed impaired entry efficiency for Omicron spike in both organoid systems and the Calu-3 cells in comparison to Delta and WT Wuhan-Hu-1 D614G. By contrast, in H1299 lung cancer epithelial cells, HeLa-ACE2 cells (overexpressing *ACE2*), and HEK293T cells overexpressing *ACE2* and deleted for *TMPRSS2* (293T-A2ΔT2), we did not observe large differences in the entry efficiency of Delta and Omicron (Fig. 3a).

To further examine our PV entry and infection findings, we studied the expression of *ACE2* and *TMPRSS2* across our target cells using qPCR analysis of RNA extracts from cell lysates (Fig. 3b). We found greater levels of *TMPRSS2* mRNA in cells in which Omicron PV entry was impaired relative to Delta—for example, Calu-3 cells and organoids had higher levels of *TMPRSS2* mRNA compared with H1299, HeLa-ACE2 and 293T-A2ΔT2 cells. *ACE2* levels were variable as expected, and did not appear to be correlated with differences in infection between Omicron

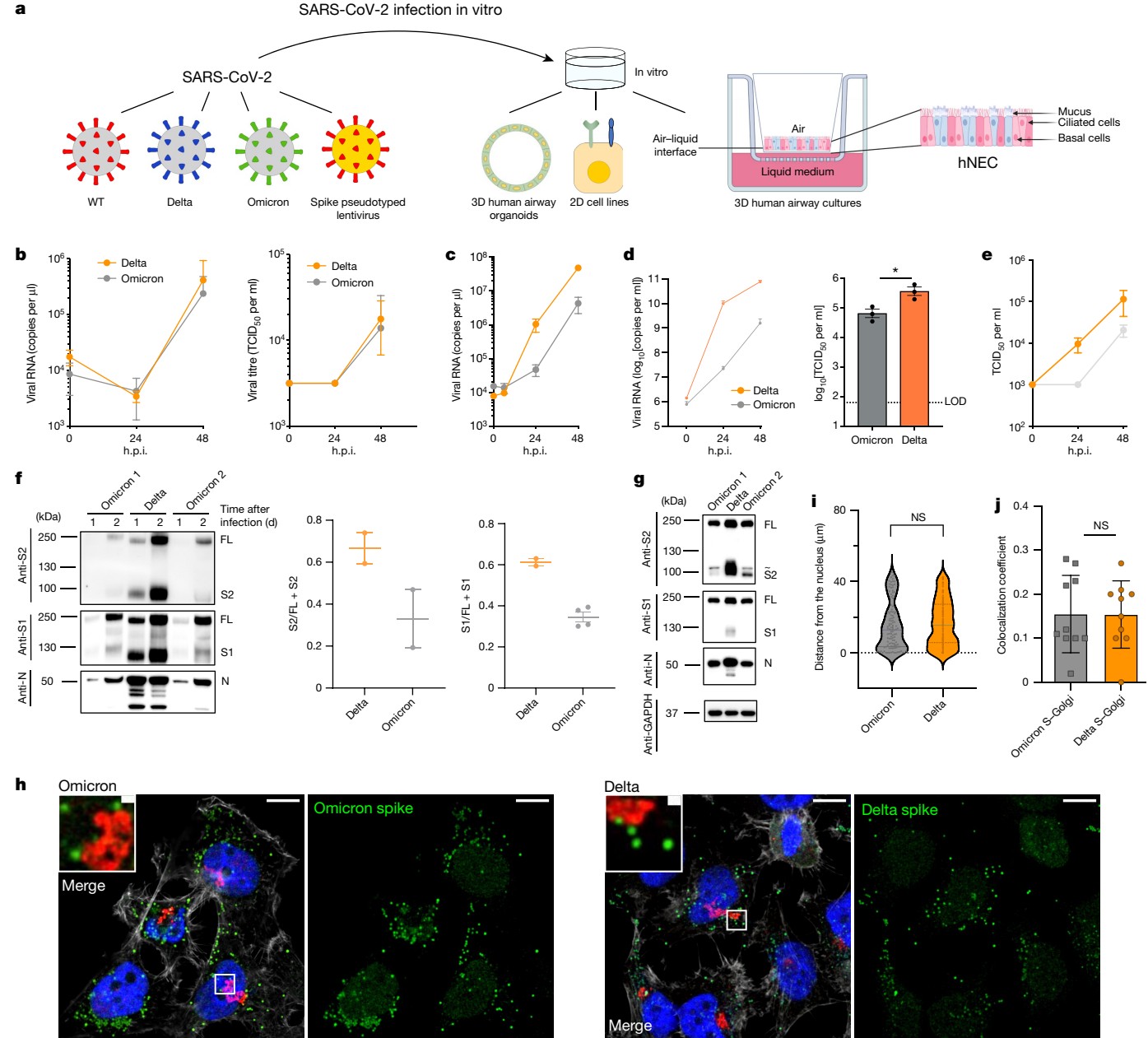

**Fig. 2 | SARS-CoV-2 Omicron and Delta Variant live virus replication in 3D tissue culture systems and 2D cell lines. a**, Overview of the viruses and culture systems used. The schematic was created using BioRender.com. **b–e**, The spread of infection by replication-competent Omicron versus Delta variants over time in hNECs (**b**), and Calu-3 (**c**), Caco-2-Npro (**d**) and HeLa-A2T2 (**e**) cells. Viral RNA and/or infectious virus in the supernatant (TCID$_{50}$) were measured. Data are mean ± s.d. of the technical replicates shown, representative of two independent experiments. Statistical analysis was performed using two-sided unpaired Student's *t*-tests; *$P < 0.05$. h.p.i., hours post-infection. **f**, Western blot analysis of two live Omicron isolates and one Delta virus isolate in Vero-ACE2/TMPRSS2 cells probed with antibodies against S2, S1 and N, with quantification of the ratio of S2 and S1 to total spike. Data are mean ± s.e.m. of 2–4 biological replicates. **g**, Vero-ACE2/TMPRSS2 producer cell lysates infected with live isolates probed with antibodies against S2, S1 and N. GAPDH was used as the loading control. A non-specific band above the S2

band is indicated (-). Data are representative of two independent experiments. **h–j**, The subcellular localization of spike in cells infected with SARS-CoV-2 Delta versus Omicron. The subcellular distribution of Omicron and Delta spike proteins in HeLa-ACE2 cells infected with live virus isolates. **h**, Cells on coverslips were infected for 24 h, fixed and stained with anti-spike, anti-GM130-cis-Golgi, phalloidin 647 and 4',6'-diamidino-2-phenylindole dihydrochloride (DAPI), and imaged on a Leica TCS SP8 confocal microscope. **i**, The distance of spike proteins from the nucleus at 24 h after infection. Data points ($n = 144$ for each virus) are shown along with the median ± interquartile range. **j**, Quantification of spike–Golgi colocalization in infected cells. Data are mean ± sd. $n = 10$ for each virus. Values were determined using Pearson's correlation coefficient. Scale bars, 10 μm (main images) and 1 μm (insets). Statistical analysis was performed using two-sided unpaired Student's *t*-tests; NS, not significant.

and Delta PV. We therefore hypothesized that the Omicron PV was disadvantaged relative to Delta in *TMPRSS2*-expressing cells.

To experimentally demonstrate differential usage of TMPRSS2 as a cofactor for virus entry by Omicron, we directly compared the efficiency of viral entry between 293T-A2ΔT2 and 293T-A2T2 cells. Enhanced infectivity was observed for both the WT and Delta when *TMPRSS2* was overexpressed, suggesting that TMPRSS2 is also required for their optimal virus entry. By contrast, no difference was observed

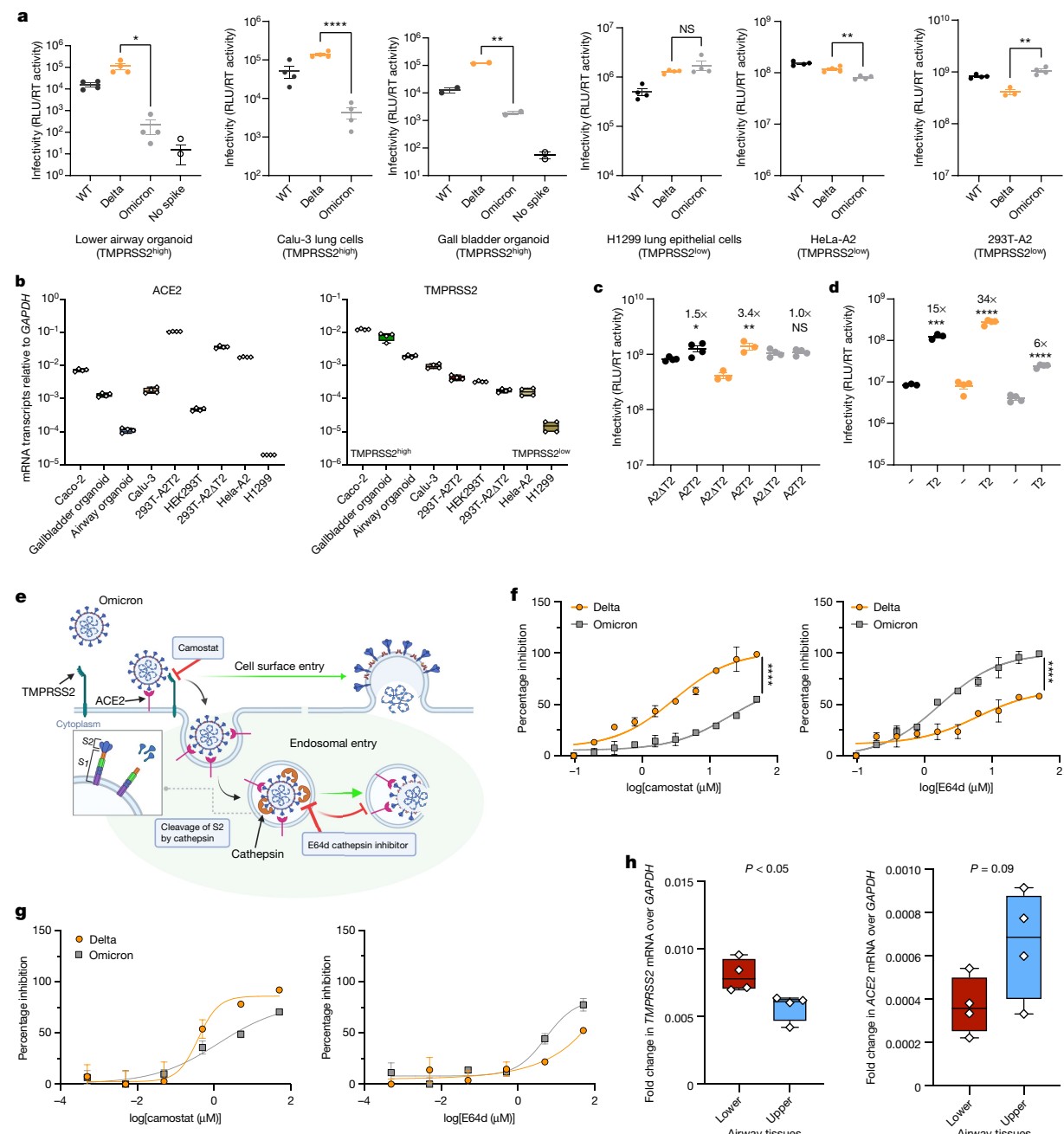

**Fig. 3 | SARS-CoV-2 Omicron variant spike enters cells less efficiently by TMPRSS2-dependent plasma membrane fusion. a**, PV entry in airway organoids, Calu-3 lung cells, gall bladded organoids, H1299 lung cells, HeLa *ACE2*-overexpressing cells and HEK293 *ACE2*-overexpressing cells. Black, WT Wuhan-Hu-1 D614G; orange, Delta/B.1.617.2; grey, Omicron/BA.1. Data are mean ± s.e.m. of $n$ = 2–4 technical replicates. RLU, relative light unit; RT, reverse transcriptase. Statistical analysis was performed using two-sided unpaired Student's *t*-tests; \**P* < 0.05, \*\**P* < 0.01, \*\*\*\**P* < 0.0001. Data are representative of three independent experiments. **b**, *ACE2* and *TMPRSS2* mRNA transcripts in the indicated cell types and organoids as measured using qPCR. Samples were run as $n$ = 4 technical replicates. The centre line shows the median, the box limits show the interquartile range and the whiskers denote the range. Data are representative of two independent experiments. **c**, Entry of PV expressing spike in HEK293T cells transduced to overexpress *ACE2* and either depleted for *TMPRSS2* (A2ΔT2) or overexpressing *TMPRSS2* (A2T2). **d**, Entry of PV expressing spike in HEK293T cells with endogenous (–) or overexpressed *TMPRSS2* (T2). For **c** and **d**, data are the mean ± s.e.m. of $n$ = 4 technical replicates, representative of three independent experiments. Statistical analysis was performed using two-sided unpaired Student's *t*-tests; \**P* < 0.05, \*\**P* < 0.01, \*\*\**P* < 0.001, \*\*\*\**P* < 0.0001. **e**, An illustration of two cell entry pathways that are known to be used by SARS-CoV-2. The schematic

was created using BioRender.com. **f**, Titration of inhibitors in A549-ACE2/ TMPRSS2 cells using PV expressing Delta (orange) or Omicron (grey) spike in the presence of the indicated doses of Camostat or E64d, then analysed after 48 h. The percentage inhibition was calculated relative to the maximum luminescence signal for each condition. For each variant and dilution, data are mean ± s.e.m. of experiments conducted in duplicate. Statistical analysis was performed using two-sided unpaired *t*-tests; \*\*\*\**P* < 0.0001. **g**, Titration of inhibitors in A549-ACE2/ TMPRSS2-based luminescent reporter cells using live virus. Cells were infected at a multiplicity of infection (m.o.i.) of 0.01 with Delta or Omicron variants in the presence of the indicated doses of Camostat or E64d, and then analysed after 24 h. The percentage inhibition was calculated relative to the maximum luminescence signal for each condition. For each variant and dilution, data are mean ± s.e.m. of experiments conducted in triplicate. Data in **f** and **g** are representative of two independent experiments. **h**, *ACE2* and *TMPRSS2* mRNA expression determined using qPCR in human lung tissue (four pieces of tissue each from upper (main bronchus) and lower (lung parenchyma) airways). Each data point is the mean of $n$ = 2 technical replicates. The centre line shows the median of biological replicates ($n$ = 4 for each lung region), the box limits show the interquartile range and the whiskers denote the range. Statistical analysis was performed using two-sided unpaired *t*-tests. No adjustments were made for multiple comparisons.

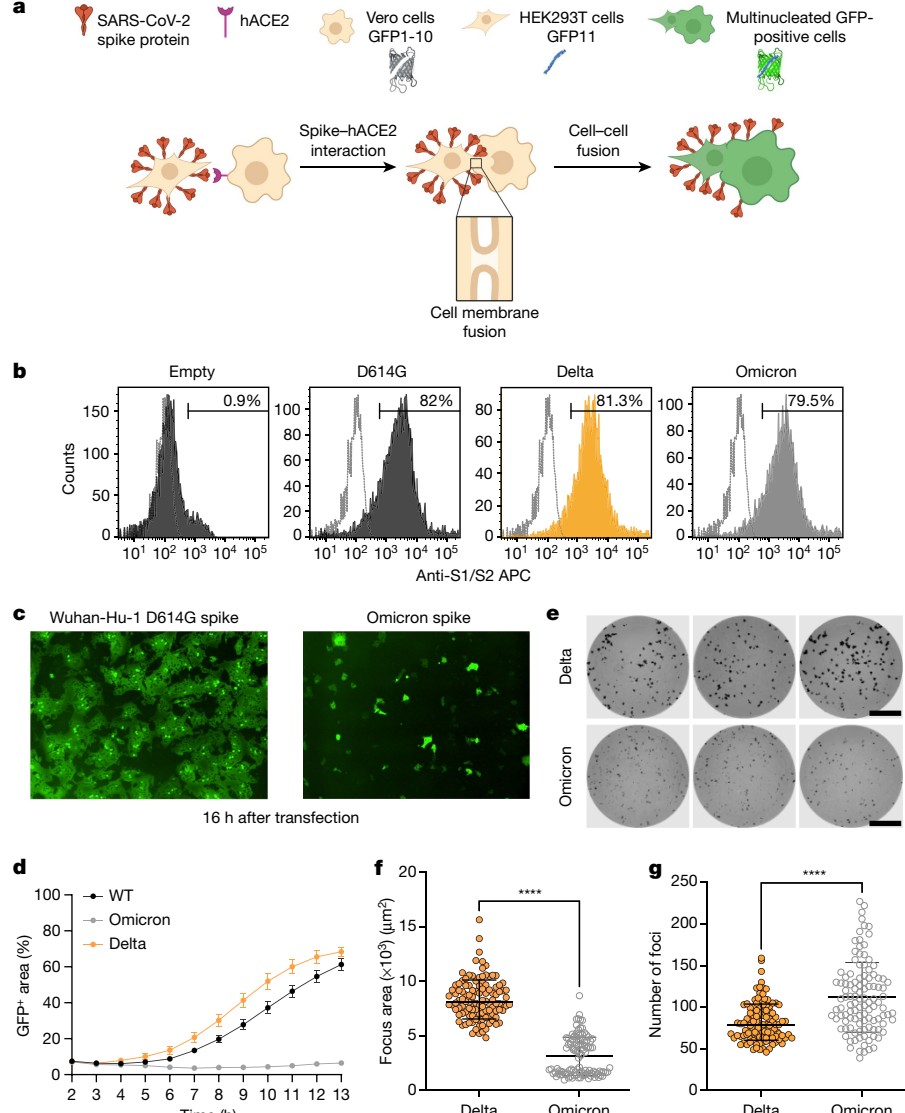

**Fig. 4 | SARS-CoV-2 Omicron variant spike shows impaired cell–cell fusion activity and smaller infection foci generated by live virus. a**, Schematic of the cell–cell fusion assay. The schematic was created using BioRender.com. **b**, Spike expression at the cell surface as determined by flow cytometry, showing the distribution of fluorescence intensity. The percentage of spike-positive cells is indicated. **c**, Reconstructed images of GFP⁺ syncytia at 16 h. **d**, Quantification of cell–cell fusion kinetics showing the percentage of the GFP⁺ area to the total cell area over time. WT, Wuhan-Hu-1 D614G. Data are mean ± s.e.m. from eight fields of views at each time point. Data are representative of at least two independent experiments. **e**, Three representative images of wells from 96-well plates with infection foci formed by Delta (top row) or Omicron (bottom row) live virus. Scale bars, 2 mm. **f**, Focus area as determined by ELISpot image analysis for Omicron (*n* = 111 wells) and Delta (*n* = 112 wells). Data are the geometric mean ± 95% CI. **g**, Focus number per well for the same experiments as in **f**. Data are the geometric mean ± 95% CI for focus number per well for Omicron and Delta infections. For **f** and **g**, statistical analysis was performed using two-sided Wilcoxon rank-sum tests; ****$P$ < 0.0001.

for Omicron PV, indicating that Omicron is inefficient in using TMPRSS2 for its entry (Fig. 3c). We hypothesized that the modest degree of TMPRSS2-dependent enhancement (3.4×) for Delta and a complete lack of enhancement for Omicron over a single infection round might relate to the overexpression of *ACE2* in this system. To examine this possibility, we tested standard HEK293T cells expressing low endogenous levels of *ACE2* and those transduced to overexpress *TMPRSS2* (293T-T2). Indeed, we observed a much greater increase in infection for Delta (34×) under TMPRSS2 overexpression conditions and a smaller increase (6×) for Omicron in the presence of overexpressed *TMPRSS2* (Fig. 3d). Together, these data support a reduced contribution of TMPRSS2 cofactor usage by Omicron spike, with modulation of the effect size by *ACE2* expression levels. Given previous data on the relationship between S1/S2 cleavage in producer cells and TMPRSS2 dependence[22,23], the most

likely explanation for altered TMPRSS2 usage is the suboptimal S1/S2 cleavage in Omicron leading to inefficient S2′ processing and fusion peptide exposure.

A change in the use of TMPRSS2 for entry would be predicted to alter the entry pathway of the virus (Fig. 3e). To test the hypothesis that the entry route preference by Omicron spike is changed, we used inhibitors of proteases specific to either the endocytic pathway (E64d blockade of cathepsins) or the plasma membrane pathway (camostat blockade of TMPRSS2). We reasoned that, if Omicron has become more dependent on the endocytic cell entry route, it should be more sensitive to cathepsin inhibition. We infected A549 cells overexpressing *ACE2* and *TMPRSS2* (A549-A2T2) cells in the presence of serial dilutions of E64d or camostat. Indeed, E64d had a greater effect on cell entry by Omicron PVs versus Delta PVs and, conversely, camostat had a greater effect on

the entry of Delta PVs (Fig. 3f). We next performed this experiment with live virus isolates, using an indicator lung cell line (A549-A2T2) with a similar, albeit less marked, result as shown for PV (Fig. 3g). These drug data in both live virus and PV systems further support the shift in tropism away from *TMPRSS2*-expressing cells.

Having established that TMPRSS2 modulates entry mediated by Delta to a greater extent than for Omicron BA.1 spike, we sought to understand the distribution of *TMPRSS2* and *ACE2* expression in human respiratory cells. We used single-nucleus RNA-sequencing (RNA-seq) data from five locations in the human lungs[24]. The comparison revealed higher expression of *TMPRSS2* in alveolar AT1 and AT2 pneumocytes and, in general, lower expression in the trachea (upper airway; Extended Data Fig. 7). *ACE2* expression was generally lower overall, but appeared to be higher in AT1 and AT2 cells compared with in other cell types, and also in club cells[25]. We experimentally extended these findings by performing qPCR analysis of human lung tissue samples[26], showing higher mRNA expression of *TMPRSS2* in the lung parenchyma (alveolar) tissue compared with in the upper-airway bronchial tissues and a trend towards *ACE2* being higher in the upper-airway tissue (Fig. 3h).

## Impaired cell–cell fusion by Omicron S

The ability of viral membrane glycoproteins to induce cell–cell fusion and syncytium formation is well established[27–29], providing an additional route for SARS-CoV-2 dissemination that may facilitate evasion of neutralizing antibodies. The role of syncytium formation in viral replication and the pathogenesis of severe COVID-19 has been reported and may be a druggable process to treat COVID-19 pathology[30,31]. SARS-CoV-2-mediated cell–cell fusion requires the PBCS and spike cleavage at S1/S2, and the process is known to be accelerated by the presence of TMPRSS2 (ref. [27]).

Mutations at P681 in the PBCS have been observed in multiple SARS-CoV-2 lineages, most notably in the B.1.1.7/Alpha variant[5] and the Delta/Kappa variants from the B.1.617 lineage[4,32]. We previously showed that spike proteins of these variants, all bearing Pro681 mutations, had significantly higher fusogenic potential compared with the D614G Wuhan-Hu-1 spike[4,33]. Omicron bears P681H, in addition to the N679K and H655Y mutations (Supplementary Fig. 6a).

Given the requirement of TMPRSS2 for optimal cell–cell fusion, we hypothesized that Omicron may be impaired in mediating this process. We used a split GFP system[34] to monitor cell–cell fusion in real time (Fig. 4a). As a control to show that spike cleavage is needed for fusion in our assay system, we titrated CMK (a furin inhibitor) into donor cell medium before transfection with spike-expressing plasmids, observing dose-dependent inhibition of fusion (Extended Data Fig. 8). As a further control to demonstrate the need for ACE2 engagement by expressed spike, we tested the ability of convalescent serum containing neutralizing antibodies to block cell–cell fusion. Indeed, serum blocked syncytia formation in a dose-dependent manner (Extended Data Fig. 8).

We performed flow cytometry to verify that spike was expressed at the cell surface (Fig. 4b). We proceeded to transfect spike-bearing plasmids into HEK293T cells expressing GFP1-10 and Vero E6 cells stably expressing GFP11, such that the GFP signal could be detected after cell–cell fusion and measured over time. (Fig. 4c, d). We observed increased fusion for Delta spike compared with D614G Wuhan-Hu-1 spike (Fig. 4d). However, the Omicron spike resulted in very poor fusion (Fig. 4d).

We predicted that poor fusion would impair cell–cell spread of Omicron, and analysed infection focus size using live virus infection of H1299 cells, in which we have shown similar entry efficiency for Omicron and Delta spike PV (Fig. 3). Infection foci in spreading virus infections occur because of localized (cell-to-cell) infection, probably facilitated by syncytium formation. The infection was followed by overlay of semi-solid medium (carboxymethylcellulose) to inhibit cell-free

infection, thereby favouring cell–cell infection. Infected cells were stained with an anti-S antibodies, and local spread was characterized by the staining of foci visible to the naked eye (Fig. 4e). Omicron infection resulted in slightly higher numbers of foci (Fig. 4e–g), but each infection focus was substantially smaller in size relative to those formed by Delta infection (Fig. 4e). For a more detailed view of the foci, we imaged at a higher magnification using a microscope objective, and used cells that were imaged 2 h after infection with ancestral virus—when relatively little viral transmission is expected to be completed—as an approximate measure of the size of a single cell (Extended Data Fig. 8). We compared this to cells that were infected for 18 h by Omicron or Delta isolates. The size of foci at 2 h after infection with ancestral virus was $1.2 \times 10^3\ \mu m^2$ (95% confidence interval (CI) = $0.9$–$1.5 \times 10^3$). The size of Omicron foci was $1.5 \times 10^3\ \mu m^2$ (95% CI = $1.4$–$1.5 \times 10^3$) and, by contrast, the size of Delta foci was $3.4 \times 10^3\ \mu m^2$ (95% CI = $3.1$–$3.7 \times 10^3$), 2.3-fold higher than Omicron (Extended Data Fig. 8). Together, these results show that localized cell–cell infection of Omicron is reduced relative to Delta, consistent with attenuated cell–cell fusion.

## Discussion

Here we examined the biological properties of Omicron from the perspectives of spike-mediated immune evasion, ACE2-binding interactions and cellular entry pathways that likely dictate tissue tropism. First, we have shown that the Omicron spike confers substantial evasion of vaccine-elicited neutralizing antibodies that is greater for ChAdOx-1 versus BNT162b2 vaccine sera. In longitudinally sampled participants, second-dose waning was mitigated by a third dose of an mRNA vaccination that also increased and broadened neutralization of Omicron in the short term. This observation is critical for informing third-dose vaccination efforts worldwide. Notably, ChAdOx-1 is widely used in low-income settings in which third doses of mRNA vaccines are not widely available, and Omicron may contribute to higher infection rates in such settings unless third doses can be implemented. Of further concern for low-income countries was the absence of any neutralizing activity to Omicron in sera after two doses of coronavac in all nine participants, in addition to poor Delta neutralization.

In terms of the implications for treatment options for moderate to severe clinical disease, we show high-level loss of in vitro activity for the widely used monoclonal antibody combination therapy REGN2 against Omicron, but no significant loss in the activity of the polymerase inhibitors remdesivir and molnupiravir against the live virus. The monoclonal antibody sotrovimab has also been reported to retain significant in vitro activity against Omicron[11,35].

Despite the presence of three mutations that are individually predicted to favour spike S1/S2 cleavage, the observed cleavage of spike in cells infected with live virus was lower compared with Delta and WT Wuhan-Hu-1 D614G. Similarly, live Omicron virions had a lower proportion of cleaved spike compared with Delta. Cleavage deficiency was also evident in PV particles. We previously noted that the proportion of cleaved spike was lower within cells as compared to virions[4], suggesting that the incorporation of spike into virions produced in the endoplasmic reticulum–Golgi intermediate compartment may favour cleaved spike over full-length spike, or be cleaved as virions are secreted through exocytic pathways[36].

Live Omicron virus showed a significant replication defect compared with Delta in cell culture systems in which TMPRSS2 was present. Omicron spike was also associated with decreased entry relative to both WT and Delta spike proteins in target lower-airway organoids, gallbladder organoids or Calu-3 lung cells expressing TMPRSS2, but displayed no difference in cells with low *TMPRSS2* expression, for example H1299, HeLa-ACE2 and 293T-A2 cells. We proceeded to directly compare entry in *ACE2*-overexpressing cells in which *TMPRSS2* was either knocked down or overexpressed, and showed that Omicron entry was not affected by variations in *TMPRSS2* expression, in contrast to WT and

Delta spike PV entry. Importantly, when *ACE2* was not overexpressed, we observed a large increase in Delta PV entry with TMPRSS2 availability, and a much smaller increase in entry for Omicron. These data suggest that TMPRSS2 cofactor usage may be affected by ACE2 levels. The hypothesis that Omicron spike has altered its preference to the cathepsin-dependent endosomal route of entry (TMPRSS2 independent) was borne out by inhibitor experiments against cathepsin and TMPRSS2 in both PV and live virus isolates.

TMPRSS2 is a member of the type II transmembrane serine protease family. This family comprises 17 members with diverse physiological functions. These proteins require activation and remain associated with membranes, and TMPRSS2 in particular can undergo autocatalytic activation[37], although its role in human physiology is unclear. TMPRSS2 has been shown to be a cofactor for SARS-CoV-1 and MERS as well as SARS-CoV-2 (refs. [3,38]). Importantly, TMPRSS2 dependence is correlated with efficient cleavage at the PBCS between S1/S2 (refs. [22,23]), with the removal of the furin-cleavage site in SARS-CoV-2 reducing TMPRSS2 dependence. By contrast, in experiments in which a furin-cleavage site was introduced into SARS-CoV-1, the virus demonstrated greater TMPRSS2 dependence.

Omicron presents an interesting parallel to SARS-CoV-1, which is not cleaved in the producer cell due to lack of the PBCS but, rather, on the target cell surface by TMPRSS2 or by cathepsins in endosomes. SARS-CoV-1 is able to infect cells though TMPRSS2 (and therefore syncytia can be induced)[29] or by endocytosis[3,23]. The SARS-CoV-2 Omicron variant appears to also favour endocytosis through reduced cleavage efficiency of spike impairing TMPRSS2-mediated entry. Indeed, the accumulated mutations in omicron include multiple additional basic amino acids leading to an increase in the overall charge of the S protein. This change may make S more sensitive to low-pH-induced conformational changes, and be an adaptation that facilitates the use of the low-pH endosomal entry route, and/or be an adaptation to entry in the lower pH environment in the upper airway. Furthermore, our analysis of single-cell RNA-seq datasets and direct qPCR measurements on human respiratory tissue samples suggest that lung cells have higher *TMPRSS2* mRNA compared with cells found in the upper airway. Indeed recent data indicate lower virus burden in deep lung tissue and reduced pathogenicity in Omicron versus Delta infections using Syrian hamster models[39].

As expected from suboptimal cleavage of spike in producer cells, and the known link between efficient spike cleavage and ability to use TMPRSS2 for cell entry/cell–cell fusion, we observed that the Omicron spike protein has low fusogenic potential. This phenomenon could translate to impaired cell–cell spread and, indeed, we observed smaller plaque sizes for Omicron compared with Delta in a system in which cell-free infection was limited by semi-solid medium. The smaller focus size for Omicron could be attributable to a failure to induce syncytia, without compromised entry efficiency given that a similar number of foci was observed. The semi-solid medium prevented cell-free infection and our measurement should therefore primarily reflect infection spread over a limited time period through syncytia formation. However, the smaller foci could have differing implications in vivo. First, there may be decreased virus production due to less cell death, therefore favouring Omicron over Delta. Alternatively, syncytia along with cell survival and larger infection foci may increase spread, thereby favouring Delta. The difference in outcome from syncytia is likely to be cell and tissue dependent and we caution against extrapolating these in vitro infection data to viral loads in the respiratory tract.

Our data showing tropism differences for Omicron in organoid systems and human nasal epithelial cultures are limited by the fact that they are in vitro systems, albeit using primary human tissue. Note also that levels of TMPRSS2 may affect ACE2, particularly as TMPRSS2 has been implicated in ACE2 cleavage[40] and our effect sizes were affected by ACE2 expression. Finally, E64d may inhibit viral protease in addition to host cellular cathepsins, and this may have affected the strength of the observed phenotype.

In summary, Omicron has gained immune evasion properties while compromising cell entry in TMPRSS2-expressing cells such as those in alveoli, as well as compromising syncytia formation—a combination of traits consistent with reduced pathogenicity in vivo. Crucially, experience with Omicron has demonstrated that predictions regarding replication and tropism based on sequence alone can be misleading, and a detailed molecular understanding of the phenotypic impact of complex sets of mutations is vital as variants continue to emerge.

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

[1]Cambridge Institute of Therapeutic Immunology & Infectious Disease (CITIID), Cambridge, UK. [2]Department of Medicine, University of Cambridge, Cambridge, UK. [3]Department of Veterinary Science, Faculty of Agriculture, University of Miyazaki, Miyazaki, Japan. [4]Division of Systems Virology, Department of Infectious Disease Control, International Research Center for Infectious Diseases, The Institute of Medical Science, The University of Tokyo, Tokyo, Japan. [5]CSIR Institute of Genomics and Integrative Biology, Delhi, India. [6]Department of Biochemistry, University of Washington, Seattle, WA, USA. [7]MRC—Laboratory of Molecular Biology, Cambridge, UK. [8]Division of Molecular Virology and Genetics, Joint Research Center for Human Retrovirus Infection, Kumamoto University, Kumamoto, Japan. [9]Division of Infection and Immunity, Joint Research Center for Human Retrovirus Infection, Kumamoto University, Kumamoto, Japan. [10]Kuramochi Clinic Interpark, Utsunomiya, Japan. [11]Human Genetics Laboratory, National Institute of Genetics, Mishima, Japan. [12]Department of Hematology and Oncology, Graduate School of Medicine, Kyoto University, Kyoto, Japan. [13]Division of Infection and Immunity, UCL, London, UK. [14]Department of Virology, University of Cambridge, Cambridge, UK. [15]Africa Health Research Institute, Durban, South Africa. [16]University of Cape Town, Cape Town, South Africa. [17]Max Planck Institute of Biochemistry, Martinsried, Germany. [18]Wellcome-MRC Cambridge Stem Cell Institute, Cambridge, UK. [19]Welcome Sanger Institute, Wellcome Trust Genome Campus, Hinxton, UK. [20]European Molecular Biology Laboratory, European Bioinformatics Institute, EMBL-EBI, Wellcome Trust Genome Campus, Hinxton, UK. [21]Addenbrooke's Hospital, Cambridge University Hospitals NHS Foundation Trust, Cambridge Biomedical Campus, Cambridge, UK. [22]Cavendish Laboratory, Department of Physics, University of Cambridge, Cambridge, UK. [23]Transplant and Regenerative Medicine Laboratory, Translational and Clinical Research Institute, Faculty of Medical Sciences, Newcastle University, Newcastle Upon Tyne, UK. [24]Humabs Biomed SA, a subsidiary of Vir Biotechnology, Bellinzona, Switzerland. [25]Department of Physiology, Development and Neuroscience, University of Cambridge, Cambridge, UK. [26]Howard Hughes Medical Institute, Seattle, WA, USA. [27]Max Planck Institute for Infection Biology, Berlin, Germany. [28]School of Laboratory Medicine and Medical Sciences, University of KwaZulu-Natal, Durban, South Africa. [29]NHS Blood and Transplant, Cambridge, UK. [30]CREST, Japan Science and Technology Agency, Saitama, Japan. [50]These authors contributed equally: Bo Meng, Adam Abdullahi, Isabella A. T. M Ferreira, Niluka Goonawardane, Akatsuki Saito, Izumi Kimura, Daichi Yamasoba. ✉e-mail: keisato@g.ecc.u-tokyo.ac.jp; rkg20@cam.ac.uk

**The CITIID-NIHR BioResource COVID-19 Collaboration**

Stephen Baker[1,2], Gordon Dougan[1,2], Christoph Hess[2], Nathalie Kingston[34,35], Paul J. Lehner[1,2], Paul A. Lyons[1,2], Nicholas J. Matheson[1,2], Willem H. Ouwehand[2], Caroline Saunders[31], Charlotte Summers[2,21], James E. D. Thaventhiran[21], Mark Toshner[2], Michael P. Weekes[2], Patrick Maxwell[21], Ashley Shaw[21], Ashlea Bucke[31], Jo Calder[31], Laura Canna[31], Jason Domingo[31], Anne Elmer[31], Stewart Fuller[31], Julie Harris[31], Sarah Hewitt[31], Jane Kennet[31], Sherly Jose[31], Jenny Kourampa[31], Anne Meadows[31], Criona O'Brien[31], Jane Price[31], Cherry Publico[31], Rebecca Rastall[31], Carla Ribeiro[31], Jane Rowlands[31], Valentina Ruffolo[31], Hugo Tordesillas[31], Ben Bullman[2], Benjamin J. Dunmore[2], Stefan Gräf[2], Josh Hodgson[2], Christopher Huang[2], Kelvin Hunter[2], Emma Jones[33], Ekaterina Legchenko[2], Cecilia Matara[2], Jennifer Martin[2], Federica Mescia[2], Ciara O'Donnell[2], Linda Pointon[2], Joy Shih[2], Rachel Sutcliffe[2], Tobias Tilly[2], Carmen Treacy[2], Zhen Tong[2], Jennifer Wood[2], Marta Wylot[2], Ariana Betancourt[2], Georgie Bower[2], Chiara Cossetti[2], Aloka De Sa[2], Madeline Epping[2], Stuart Fawke[2], Nick Gleadall[2], Richard Grenfell[2], Andrew Hinch[2], Sarah Jackson[2], Isobel Jarvis[21], Ben Krishna[2], Francesca Nice[31], Ommar Omarjee[2], Marianne Perera[2], Martin Potts[2], Nathan Richoz[2], Veronika Romashova[2], Luca Stefanucci[2], Mateusz Strezlecki[2], Lori Turner[2], Eckart M. D. D. De Bie[2], Katherine Bunclark[2], Masa Josipovic[2], Michael Mackay[2], Helen Butcher[34,36], Daniela Caputo[34,36], Matt Chandler[34,36], Patrick Chinnery[34,38,39], Debbie Clapham-Riley[34,36], Eleanor Dewhurst[34,36], Christian Fernandez[34,35], Anita Furlong[34,36], Barbara Graves[34,36], Jennifer Gray[34,36], Sabine Hein[34,36], Tasmin Ivers[34,36], Emma Le Gresley[34,36], Rachel Linger[34,36], Mary Kasanicki[21,34], Rebecca King[34,36], Nathalie Kingston[34,35], Sarah Meloy[34,36], Alexei Moulton[34,36], Francesca Muldoon[34,36], Nigel Ovington[34,35], Sofia Papadia[34,36], Christopher J. Penkett[34,35], Isabel Phelan[34,36], Venkatesh Ranganath[34,35], Roxana Paraschiv[34,35], Abigail Sage[34,36], Jennifer Sambrook[34,35], Ingrid Scholtes[34,36], Katherine Schon[34,37,38], Hannah Stark[34,36], Kathleen E. Stirrups[34,35], Paul Townsend[34,35], Neil Walker[34,35] & Jennifer Webster[34,36]

[31]Cambridge Clinical Research Centre, NIHR Clinical Research Facility, Cambridge University Hospitals NHS Foundation Trust, Addenbrooke's Hospital, Cambridge, UK. [33]Universidad San Francisco de Quito, COCIBA, Laboratorio de Biotecnología Vegetal, Cumbaya, Ecuador. [34]NIHR BioResource, Cambridge University Hospitals NHS Foundation Trust, Cambridge Biomedical Campus, Cambridge, UK. [35]Department of Haematology, School of Clinical Medicine, University of Cambridge, Cambridge Biomedical Campus, Cambridge, UK. [36]Department of Public Health and Primary Care, School of Clinical Medicine, University of Cambridge, Cambridge Biomedical Campus, Cambridge, UK. [37]Clinical Genetics, Addenbrooke's Hospital, Cambridge University Hospitals NHS Foundation Trust, Cambridge, UK. [38]Department of Clinical Neurosciences, School of Clinical Medicine, University of Cambridge, Cambridge Biomedical Campus, Cambridge, UK. [39]Medical Research Council Mitochondrial Biology Unit, Cambridge Biomedical Campus, Cambridge, UK.

**The Genotype to Phenotype Japan (G2P-Japan) Consortium**

Erika P. Butlertanaka[3], Yuri L. Tanaka[3], Jumpei Ito[4], Izumi Kimura[4,50], Keiya Uriu[4], Yusuke Kosugi[4], Mai Suganami[4], Akiko Oide[4], Miyabishara Yokoyama[4], Mika Chiba[4], Chihiro Motozono[9], Hesham Nasser[8], Ryo Shimizu[8], Kazuko Kitazato[8], Haruyo Hasebe[8], Takashi Irie[40], So Nakagawa[41], Jiaqi Wu[41], Miyoko Takahashi[41], Takasuke Fukuhara[42], Kenta Shimizu[42], Kana Tsushima[42], Haruko Kubo[42], Yasuhiro Kazuma[12], Ryosuke Nomura[12], Yoshihito Horisawa[12], Kayoko Nagata[12], Yugo Kawai[12], Yohei Yanagida[12], Yusuke Tashiro[12], Kenzo Tokunaga[32], Seiya Ozono[32], Ryoko Kawabata[40], Nanami Morizako[3], Kenji Sadamasu[43], Hiroyuki Asakura[43], Mami Nagashima[43] & Kazuhisa Yoshimura[43]

[32]National Institute of Infectious Diseases, Tokyo, Japan. [40]Hiroshima University, Hiroshima, Japan. [41]Tokai University, Kanagawa, Japan. [42]Hokkaido University, Sapporo, Japan. [43]Tokyo Metropolitan Institute of Public Health, Tokyo, Japan.

**Ecuador-COVID19 Consortium**

Paúl Cárdenas[44], Erika Muñoz[44], Veronica Barragan[44], Sully Márquez[44], Belén Prado-Vivar[44], Mónica Becerra-Wong[44], Mateo Caravajal[44], Gabriel Trueba[44], Patricio Rojas-Silva[44], Michelle Grunauer[45], Bernardo Gutierrez[33], Juan José Guadalupe[33], Juan Carlos Fernández-Cadena[46], Derly Andrade-Molina[47], Manuel Baldeon[48] & Andrea Pinos[49]

[44]Universidad San Francisco de Quito, COCIBA, Instituto de Microbiología, Cumbaya, Ecuador. [45]Universidad San Francisco de Quito, COCSA, Escuela de Medicina, Cumbaya, Ecuador. [46]Laboratorio INTERLAB, Guayaquil, Ecuador. [47]Universidad Espíritu Santo, Laboratorio de Omicas, Guayaquil, Ecuador. [48]Universidad Internacional del Ecuador, Facultad de Ciencias Médicas, de la Salud y la Vida, Quito, Ecuador. [49]Centros Médicos Dr Marco Albuja, Quito, Ecuador.
.

# Methods

## Serum and tissue samples and ethical approval

Ethical approval for study of vaccine-elicited antibodies in sera from vaccinated individuals was obtained from the East of England–Cambridge Central Research Ethics Committee Cambridge (REC ref: 17/EE/0025). The participants provided informed consent. The samples of human upper and lower airways were obtained from the lungs of a multiorgan donor whose lungs were deemed to be unsuitable for clinical transplantation and after their next of kin consented to their use in research. The studies using human donor lung tissue were approved by National Research Ethics Committee (NREC) 16/NE/0230.

All protocols involving samples from human participants recruited at Kyoto University, Kuramochi Clinic Interpark and Universidad San Francisco de Quito were reviewed and approved by the Institutional Review Boards of Kyoto University (approval ID: G0697), Kuramochi Clinic Interpark (approval ID: G2021-004) and Universidad San Francisco de Quito (approval ID: CEISH P2020-022IN) and the Ecuadorian Ministry of Health (approval IDs: MSP-CGDES-2020-0121-O and MSP-CGDES-061-2020). The export of sera from Ecuador to Japan was approved by ARCSA ID: ARCSA-ARCSA-CGTC-DTRSNSOYA-2021-1626-M. All human participants provided written informed consent. All protocols for the use of human specimens were reviewed and approved by the Institutional Review Boards of The Institute of Medical Science, The University of Tokyo (approval ID: 2021-1-0416), Kumamoto University (approval IDs: 2066 and 2074) and University of Miyazaki (approval ID: O-1021)

## Omicron spike model showing surface mutations with annotation

Spike genomes for the original Wuhan strain, and Omicron VOC were obtained from GISAID EpiCoV database accessed on 30th November 2021. A consensus Spike genome was created from all complete and high-coverage genomes, excluding all sequences with >5% Ns using Geneious Prime v.2022. The consensus genome was translated to poly-proteins and the spike gene was aligned to the original Wuhan strain using mafft (v.7.490)[41] with the parameters --globalpair --maxiterate 1000 flags.

Alternatively, 3D structural models of the spike homotrimer protein complex were generated using Alphafold (v.2.1.1)[42]. In its validation at the 14th edition of the Critical Assessment of protein Structure Prediction (CASP14), the predictions generated were demonstrated to be comparative to experimental structures. When a close homologue structure is available to Alphafold, the predictions that it generates for those positions are within typical experimental error. The required databases were downloaded on 2 December 2021. The program was run with the parameters --max_template_date=2021-12-01 --model_preset=monomer --db_preset=full_dbs --is_prokaryote_list=false. Protein structures were visualized in ChimeraX (v.1.3)[43]. As predicted structures for the whole spike protein include poorly resolved chains at the terminal ends, these residues were identified by overlaying structures on Protein Data Bank (PDB) entry 6ZP2, then selected and removed from PDB files using the delete atoms/bonds action. Two further monomers were overlayed on 6ZP2 to generate a homotrimer structure. Mutated residues were then coloured in red and labelled manually with respect to the Wuhan strain.

## Spike model comparing Delta and Omicron

The SARS-CoV-2 Delta (B.1.617.2) and Omicron (B.1.529) models in Extended Data Fig. 1a were generated using the cryo-EM structure PDB 7A94 as a reference. The experimental structure has been resolved in open conformational state with one RBD in the 'up' conformation bound to ACE2. To generate the complete structure, the missing loop regions in the structure (71–75, 625–632, 677–688, 828–852 and 941–943) were modelled using the Modeller plug-in[2]. The hACE2 structure in

the template also had six residues missing patches that were added by using PDB 1R42 as a reference[3]. The top model was selected on the basis of the zDOPE score and the least root-mean-square deviation (r.m.s.d.) from the reference structure. To create variant models, mutations were induced using the Dunbrack 2010 backbone-dependent rotamer library in Chimera[4]. Further processing of steps involving minimization and peptide bond building (after deletions for B.1.617.2 and B.1.529 spike) were performed using the Gromacs. The resultant Delta and Omicron modelled complexes have 3,988 and 3,994 residues, respectively, with spike trimer consisting of three chains (14–1,146 residues). The structural superimposition of the Delta and Omicron spike model with the cryo-EM structure displayed an overall r.m.s.d. of 0.66 Å and 0.87 Å, respectively. All of the molecular images were rendered using ChimeraX[5].

## Biolayer interferometry analysis

Assays were performed on an Octet Red (ForteBio) instrument at 30 °C with shaking at 1,000 rpm. Streptavidin (SA) biosensors were hydrated in water for 10 min before a 1 min incubation in undiluted 10× kinetics buffer. SARS-CoV-2 Wuhan, Delta or Omicron RBDs were loaded at 2.5 μg ml$^{-1}$ in 10× kinetics buffer for 100–150 s before baseline equilibration for 120 s in 10× kinetics buffer. Human ACE2 association with Wuhan, Delta and Omicron RBDs was assessed in 10× kinetics buffer at various concentrations in a threefold dilution series from 333.3 to 4.1 nM and carried out for 300 s before dissociation for 300 s. The data were normalized to the baseline and fitting was performed using a 1:1 binding model and the ForteBio data analysis software. Kinetics values ($K_D$, $k_{on}$, $k_{off}$) were determined with a global fit applied to all data.

The SARS-CoV-2-RBD-Avi constructs were synthesized by GenScript into pcDNA3.1- with an N-terminal mu-phosphatase signal peptide and a C-terminal octa-histidine tag, flexible linker and avi tag (GHHHHHHHHGGSSGLNDIFEAQKIEWHE). The boundaries of the construct are N-328RFPN331 and 528KKST531-C[10,13]. The human angiotensin-converting enzyme ectodomain (hACE2) consists of residues 1–614 fused to a C-terminal octa-histidine tag[12]. Proteins were produced in Expi293F Cells (Thermo Fisher Scientific) grown in suspension using Expi293 Expression Medium (Thermo Fisher Scientific) at 37 °C in a humidified 8% $CO_2$ incubator rotating at 130 rpm. Cells grown to a density of 3 million cells per ml were transfected using the ExpiFectamine 293 Transfection Kit (Thermo Fisher Scientific) and cultivated for 4 days. Proteins were purified from clarified supernatants using a nickel HisTrap HP affinity column (Cytiva) and washed with ten column volumes of 20 mM imidazole, 25 mM sodium phosphate pH 8.0 and 300 mM NaCl before elution on a gradient to 500 mM imidazole. Proteins were buffer-exchanged into 20 mM sodium phosphate pH 8 and 100 mM NaCl and concentrated using centrifugal filters (Amicon Ultra) before being flash-frozen.

## Flow cytometry analysis of spike protein surface expression

ExpiCHO cells were seeded into 50 ml Mini Bioreactor Centrifuge Tube (Corning) at $6 \times 10^6$ cells per ml in 5 ml ExpiCHO Expression medium (Life technology). Plasmids encoding SARS-CoV-2 Wuhan, Alpha (B.1.1.7), Delta (B.1.617.2) or Omicron (B.1.1.529) spike protein (5 mg) were diluted in OptiPRO SFM (Life Technology) and mixed with Expifectamine CHO Reagent (Life Technology). After incubation for 1 min at room temperature, transfection mixes were added to the cell suspensions. Next, the cells were incubated at 37 °C under 8% $CO_2$ with an orbital shaking speed of 120 rpm for the next 48 h.

Transiently transfected cells were collected and washed with PBS, 1% BSA and 2 mM EDTA. Cells were counted, dispensed into round-bottom 96-well plates (Corning) and incubated with human IgG Fc-conjugated ACE2 serial dilutions (concentration range: 30,000–0.17 ng ml$^{-1}$) for 45 min at 4 °C. After two washes, Alexa Fluor647 Goat Anti-Human IgG secondary antibodies (1.5 mg ml$^{-1}$) (Jackson Immunoresearch) was added to the cells, which were then incubated for

30 min at 4 °C in the dark. The cells were washed twice and resuspended in wash buffer for data acquisition at ZE5 cytometer (Bio-Rad). To assess spike protein expression level, an aliquot of each transfectant cell line was stained with 10 mg ml$^{-1}$ of S2P6 antibody[35] for 45 min at 4 °C.

### Analysis of single-nucleus and single-cell RNA-seq datasets
For the tissue data[24], only single-nucleus RNA-seq were selected. Scanpy v.1.7.1 was used to process the data. ACE2 and TMPRSS2 expression was plotted using the violin plot function in scanpy.

### Cells and cell culture
Caco-2 cells transduced with BVDV NPro (Caco-NPro) were cultured in Dulbecco's modified Eagle's medium (DMEM) supplemented with 10% FCS, non-essential amino acid (NEAA), glutamine and 100 U ml$^{-1}$ penicillin and 100 μg ml$^{-1}$ streptomycin at 37 °C with 5% CO$_2$. HOS-ACE2/TMPRSS2 cells (HOS cells stably expressing human ACE2 and TMPRESS2)[6,44] were maintained in Dulbecco's modified Eagle's medium (high glucose) (Wako, 044-29765) containing 10% foetal bovine serum (FBS) and 1% penicillin–streptomycin. VeroE6/TMPRSS2 cells (an African green monkey (*Chlorocebus sabaeus*) kidney cell line; JCRB1819)[45] were maintained in Dulbecco's modified Eagle's medium (low glucose) (Wako, 041-29775) containing 10% FBS, G418 (1 mg ml$^{-1}$; Nacalai Tesque, G8168-10ML) and 1% penicillin–streptomycin. Calu-3 (a human lung epithelial cell line; a gift from P. Lehner, or ATCC HTB-55) were maintained in Eagle's minimum essential medium (Sigma-Aldrich, M4655-500ML) containing 10% FBS and 1% penicillin–streptomycin. Vero E6 (ATCC-CRL1586) and Vero-ACE2/TMPRSS2 (a gift from E. Thomson) were maintained in Dulbecco's modified Eagle's medium (low glucose) (Wako, 041-29775) containing 10% FBS, G418 (1 mg ml$^{-1}$; Nacalai Tesque, G8168-10ML) and 1% penicillin–streptomycin. HeLa-ACE2 (a gift from J. Voss), A549-ACE2/TMPRSS2 (a gift from M. Palmarini) and HeLa-ACE2/TMPRSS2 were maintained in DMEM containing 10% FBS and 1% penicillin–streptomycin. HEK293T (CRL-3216) cells, and their derivative cell lines, including 293T-ACE2/TMPRSS2, 293T-ACE2ΔTMPRSS2, 293T-TMPRSS2 and 293T-GFP11, have been described previously[22] and were provided by L. James. All the HEK293T cell lines as well as Vero-GFP1-10 (also a gift from L. James) were maintained in DMEM with 10% FBS and 1% penicillin–streptomycin.

H1299 cells (a gift from S. Cook) were maintained in Roswell Park Memorial Institute (RPMI) medium supplemented with 10% FCS and 1% penicillin–streptomycin. All of the cells were regularly tested and are mycoplasma free. Airway epithelial organoids and gallbladder organoids were obtained and maintained as previously described[20,26]. For airway organoids, human distal lung parenchymal tissues were obtained from adult donors with no background lung pathologies from Papworth Hospital Research Tissue Bank (T02233). Airway organoids were cultured in 48-well plate were passaged every 2 weeks. For passaging, Matrigel was melted by incubation with dispase at 37 °C for 30 min, followed by single-cell dissociation by TrypLE (Gibco) at 37 °C for 5 min. Then, 5,000 cells were resuspended in fresh GFR-Matrigel, followed by plating in 48-well plates. Primary cholangiocyte organoids were isolated from primary biliary tissue (intrahepatic ducts, common bile duct and gallbladder) and cultured using established methodology as previously reported[26].

### Virus isolates
The lineages B.1.617.2 (Delta, GISAID: EPI_ISL_1731019) and B.1.1.529 (ref. [46]) (Omicron UK isolate[47], G. Screaton) were received as part of the work conducted by G2P-UK National Virology Consortium (Barclay PI). Virus was propagated on Vero-ACE2-TMPRSS2 (VAT) cells for 3 days until cytopathic effect (CPE) was observed. These viruses were sequence-confirmed and were used for western blotting and Caco-3 gut experiments.

A second Omicron isolate from South Africa was obtained (GISAID: EPI_ISL_7886688), and used specifically as a second isolate for western blot analysis of live virions and for focus-size experiments. To isolate virus, ACE2-expressing H1299 cells were seeded at $4.5 \times 10^5$ cells in a six-well plate well and incubated for 18–20 h. After one DPBS wash, the sub-confluent cell monolayer was inoculated with 500 μl universal transport medium diluted 1:1 with growth medium filtered through a 0.45 μm filter. Cells were incubated for 1 h. Wells were then filled with 3 ml complete growth medium. After 4 days of infection (completion of passage 1 (P1)), cells were trypsinized, centrifuged at 300 rcf for 3 min and resuspended in 4 ml growth medium. All of the infected cells were then added to Vero E6 cells that had been seeded at $2 \times 10^5$ cells per ml, 20 ml total, 18–20 h earlier in a T75 flask for cell-to-cell infection. The coculture of ACE2-expressing H1299-E3 (clone 3) and Vero E6 cells was incubated for 1 h and the flask was then filled with 20 ml of complete growth medium and incubated for 4 days. The viral supernatant (passage 2 (P2) stock) was used for experiments. P2 stock was sequenced and confirmed to be Omicron BA.1. The sequence was deposited in GISAID under accession number EPI_ISL_7886688.

A third Omicron isolate (BA.1 lineage, strain TY38-873; GISAID: EPI_ISL_7418017) with same mutational profile as other Omicron variant isolates and provided by the National Institute for Infectious Diseases, Japan, was used for hNEC, Calu-3 and HeLa-ACE2-TMPRSS2 infection experiments. Virus isolation was performed as previously described[45]. In brief, the saliva was inoculated into VeroE6/TMPRSS2 cells and CPE was observed 4 days after inoculation. The supernatant was then collected and stored at −80 °C as an original virus (GISAID: EPI_ISL_7418017). After one more passage in VeroE6/TMPRSS2 cells, the following virus was additionally obtained from National Institute of Infectious Diseases, Japan for hNEC experiments: Delta isolate (B.1.617.2 lineage, strain TKYTK1734; GISAID: EPI_ISL_2378732)[6]. Virus preparation and titration of virus isolates was performed as previously described[6,44]. To prepare the working virus stock, 100 μl of the seed virus was inoculated into VeroE6/TMPRSS2 cells ($5 \times 10^6$ cells in a T-75 flask). In brief, 1 h after infection, the culture medium was replaced with fresh culture medium. At 3 days after infection, the culture medium was collected and centrifuged, and the supernatants were collected as the working virus stock. The titre of the prepared working virus was measured as the TCID$_{50}$ in VeroE6/TMPRSS2 cells. Serially diluted virus stocks were inoculated into the cells and incubated at 37 °C for 4 days. The cells were observed under microscopy to judge the CPE appearance. The value of TCID$_{50}$ per ml was calculated using the Reed–Muench method.

### Serum collection for live-virus neutralization experiments
Vaccine sera were collected from 14 vaccinated individuals 4 weeks after the second vaccination with BNT162b2 (Pfizer-BioNTech) (average age: 46, range: 38–55, 21% male). The sera obtained from 7 vaccinated individuals 2–3 weeks after the second vaccination with ChAdOx1 (Oxford-AstraZeneca) (average age: 46, range: 35–54, 71% male) were purchased from BioIVT.

Convalescent sera were collected from ten vaccine-naive individuals who had been infected with Delta variant (B.1.617.2 or AY.29) (average age: 46, range: 22–63, 70% male). To determine the SARS-CoV-2 variants that individuals were infected with, saliva samples were collected from patients with COVID-19 during onset and RNA was extracted using the QIAamp viral RNA mini kit (Qiagen, Qiagen, 52906) according to the manufacturer's protocol. The sample was analysed using whole-genome sequencing based on a modified ARTIC Network protocol[48], and the near-full-length SARS-CoV-2 genome sequences were obtained. Alternatively, we also performed the capture-hybridization method. The RNA isolated from saliva was treated with DNase I (Takara, EN0521) and the sequencing library was prepared using the Twist library preparation kit (Twist Bioscience, 101058). The capture-hybridization was conducted using xGen COVID-19 capture panel and xGen hybridization and wash kit (Integrated DNA Technologies, 1080578) according to the manufacturer's instruction. Illumina sequencing was performed using the MiSeq reagent kit v2 (300 cycles) and a MiSeq sequencer (Illumina).

For the data analysis, trimmomatic-0.39 (ref. [49]) was used to remove the adapters and low-quality reads from the raw sequencing data. The trimmed paired-end reads were aligned to the human genome hg38 using bowtie2 (v.2.3.4.3)[50] and unmapped reads were mapped to the original SARS-CoV-2 genome (strain Wuhan-Hu-1, GenBank accession no. NC_045512.2) using Bwa-mem2 (https://github.com/bwa-mem2/bwa-mem2). The PCR duplicates were removed by gencore (v.0.16.0)[51] and a consensus sequence was obtained using IGV (v.2.10.2)[52]. The mutations detected and viral lineage were determined by using CoVsurver (https://corona.bii.a-star.edu.sg) and Pangolin COVID-19 lineage assigner (https://pangolin.cog-uk.io/). The twelve convalescent sera during early pandemic (until April 2020) (average age: 71, range: 52–92, 8% male) were purchased from RayBiotech. Sera were inactivated at 56 °C for 30 min and stored at −80 °C until use.

## hNEC infection
Primary human nasal epithelial cells (EP02, MP0010) were purchased from Epithelix and maintained according to the manufacturer's procedure. The infection experiment using primary human nasal epithelial cells was performed as follows. In brief, the working Omicron (EPI_ISL_7418017) and Delta (EPI_ISL_2378732) viruses were diluted with Opti-MEM (Thermo Fisher Scientific, 11058021). The diluted viruses (1,000 TCID$_{50}$ in 100 μl) were inoculated onto the apical side of the culture and incubated at 37 °C for 1 h. The inoculated viruses were removed and washed twice with Opti-MEM. To collect the viruses on the apical side of the culture, 100 μl Opti-MEM was applied onto the apical side of the culture and incubated at 37 °C for 10 min. The Opti-MEM applied was collected and analysed using qPCR with reverse transcription (RT–qPCR) to quantify the viral RNA copy number (see below).

## Caco-2 gut cell infection
Caco-2-Npro cells were prepared in 48-well plates and infected in biological triplicates with either SARS-CoV-2 Delta (EPI_ISL_1731019) or Omicron (M21021166) variant at a m.o.i. of 1 or 0.1 TCID$_{50}$ per cell in 250 μl DMEM containing 2% FBS for 2 h at 37 °C with 5% $CO_2$. Unabsorbed inoculum was then removed, cells were washed and the medium was replaced with DMEM containing 2% FBS. Virus infections were maintained up to 72 h. Cell culture supernatants were collected at 0, 24 and 48 h after infection, and viral RNAs were extracted and quantified using RT–qPCR using specific primer probe against the *E* gene. An absolute quantification against a standard curve generated was carried out using in vitro-transcribed RNA from fragment 11 clone provided by V. Thiel. Data were collected using the ViiA 7 Real-Time PCR System (Applied Biosystems). Moreover, tenfold serial dilutions of collected virus supernatants were prepared in DMEM culture medium for measuring the infectious dose by TCID$_{50}$. Virus titres were recorded at 72 h after infection for CPE appearance and expressed as TCID$_{50}$ per ml values using the Reed–Muench method .

## Calu-3 lung cell and HeLa-ACE2-TMPRSS2 live virus infection
One day before infection, Calu-3 cells (20,000 cells) and HeLa-ACE2/TMPRSS2 cells (10,000 cells) were seeded onto a 96-well plate. SARS-CoV-2 (100 TCID$_{50}$ for HeLa-ACE2/TMPRSS2 cells; and 2,000 TCID$_{50}$ for Calu-3 cells) Omicron (EPI_ISL_7418017) or Delta (EPI_ISL_1731019) was inoculated and incubated at 37 °C for 1 h. The infected cells were washed, and 180 μl of culture medium was added. The culture supernatant (10 μl) was collected at the indicated time points and used for real-time RT–PCR to quantify the viral RNA copy number[6,44]. In brief, 5 μl of culture supernatant was mixed with 5 μl of 2× RNA lysis buffer (2% Triton X-100, 50 mM KCl, 100 mM Tris-HCl (pH 7.4), 40% glycerol, 0.8 U μl$^{-1}$ recombinant RNase inhibitor (Takara, 2313B)) and incubated at room temperature for 10 min. RNase-free water (90 μl) was added, and the diluted sample (2.5 μl) was used as the template for RT–qPCR performed according to the manufacturer's protocol using the One Step TB Green PrimeScript PLUS RT-PCR kit (Takara, RR096A) and

the following primers: forward *N*, 5′-AGC CTC TTC TCG TTC CTC ATC AC-3′; and Reverse *N*, 5′-CCG CCA TTG CCA GCC ATT C-3′. The viral RNA copy number was standardized using the SARS-CoV-2 direct detection RT-qPCR kit (Takara, RC300A). Fluorescent signals were acquired using the QuantStudio 3 Real-Time PCR system (Thermo Fisher Scientific), a CFX Connect Real-Time PCR Detection system (Bio-Rad), an Eco Real-Time PCR System (Illumina) or a 7500 Real Time PCR System (Applied Biosystems).

## Western blot analysis of infected cells and virions
VeroE6-ACE2/TMPRSS2 cells were infected with Delta (EPI_ISL_1731019), Omicron isolate 1 (M21021166) and 2 (EPI_ISL_7886688) at an m.o.i. of 1. At 24 and 48 h of infection, cells and culture medium were collected. The culture media were centrifuged, and the supernatants were collected. To pellet down SARS-CoV-2 in the supernatants, an equal volume of clarified supernatants was mixed with 20% PEG6000 in PBS and centrifuged at 12,000*g* for 30 min at 4 °C (ref. [22]), followed by pellet resuspension in 1× SDS sample buffer.

For cell lysates, the collected cells were washed and lysed in lysis buffer (Cell Signalling) and the lysates were diluted with 4 × sample buffer (Bio-Rad) and boiled for 10 min before analysed using western blotting. For protein detection, the following antibodies were used: rabbit anti-SARS-CoV-2 S monoclonal antibodies (PA1-41165, Thermo Fisher Scientific), mouse anti-SARS-CoV-2 S1 antibodies (MAB105403, R&D systems), rabbit anti-SARS-CoV-2 N monoclonal antibodies (HL344, GeneTex, GTX635679), rabbit anti-GAPDH polyclonal antibodies (10494-1-AP, Proteintech), horseradish peroxidase (HRP)-conjugated anti-rabbit and anti-mouse IgG polyclonal antibodies (Cell Signalling). Chemiluminescence was detected using the ChemiDoc Touch Imaging System (Bio-Rad). The cleavage ratio of S1 or S2 to FL in virions was determined by densitometry using ImageJ (NIH)

## Pseudotype virus preparation for testing against vaccine-elicited antibodies and cell entry
Plasmids encoding the spike protein of SARS-CoV-2 D614 with a C terminal 19 amino acid deletion with D614G were used. Omicron and Delta spikes were generated by gene synthesis. Viral vectors were prepared by transfection of HEK293T cells using the Fugene HD transfection reagent (Promega). HEK293T cells were transfected with a mixture of 11 ul of Fugene HD, 1 μg of pCDNAΔ19 spike, 1 ug of p8.91 HIV-1 gag-pol expression vector and 1.5 μg of pCSFLW (expressing the firefly luciferase reporter gene with the HIV-1 packaging signal). Viral supernatant was collected at 48 and 72 h after transfection, filtered through 0.45 um filter and stored at −80 °C as previously described. Infectivity was measured by luciferase detection in target cells.

## Neutralization titre analyses
Neutralization by vaccine-elicited antibodies after two doses of the BNT162b2 and Chad-Ox-1 vaccine, in addition to a third dose with BNT162b2, was determined by PV infections in the presence of serial dilutions of sera. The ID$_{50}$ values within groups were summarized as a geometric mean titre and statistical comparisons between groups were performed using Mann–Whitney *U*-tests or Wilcoxon signed-rank tests. Statistical analyses were performed using Stata v.13 and Prism v.9.

## Preparation of monoclonal antibodies
Casirivimab and Imdevimab were prepared as previously described[53]. To construct the plasmids expressing anti-SARS-CoV-2 monoclonal antibodies (casirivimab and imdevimab), the sequences of the variable regions of casirivimab and imdevimab were obtained from KEGG Drug Database (https://www.genome.jp/kegg/drug/) and were artificially synthesized (Fasmac). The obtained coding sequences of the variable regions of the heavy and light chains were cloned into the pCAGGS vector containing the sequences of the human immunoglobulin 1 and κ constant region (provided by H. Arase).

To prepare these monoclonal antibodies, the pCAGGS vectors containing the sequences encoding the immunoglobulin heavy and light chains were cotransfected into HEK293T cells using PEI Max (Polysciences, 24765-1). At 48 h after transfection, the cell culture supernatants were collected, and the antibodies were purified using the NAb protein A plus spin kit (Thermo Fisher Scientific, 89948) according to the manufacturer's protocol.

## Neutralization assay for monoclonal antibodies with live virus

One day before infection, VeroE6/TMPRSS2 (10,000 cells) were seeded into a 96-well plate. The monoclonal antibodies (casirivimab, imdevimab or casirivimab/imdevimab) and the heat-inactivated human sera were serially diluted with DMEM supplemented with 10% FCS and 1% penicillin–streptomycin. The diluted antibodies and sera were incubated with SARS-CoV-2 (120 $TCID_{50}$) Omicron (EPI_ISL_7418017) or Delta (EPI_ISL_2378732) isolates at 37 °C for 1 h. The viruses without antibodies or sera were included as controls. The mixture (containing the virus at 100 $TCID_{50}$) was inoculated onto a monolayer of VeroE6/TMPRSS2 cells and incubated at 37 °C for 1 h. The cells were next washed with DMEM and cultured in DMEM supplemented with 10% FCS and 1% penicillin–streptomycin. At 24 h after infection, the culture supernatants were collected and viral RNA was quantified using RT–qPCR. The assay of each antibody or serum was performed in triplicate or quadruplicate, and the 50% neutralization titre was calculated using Prism 9 (GraphPad).

## Antiviral drug assay with live virus

One day before infection, HOS-ACE2-TMPRSS2 cells (10,000 cells) were seeded into a 96-well plate. The cells were infected with SARS-CoV-2 (100 $TCID_{50}$) Omicron (EPI_ISL_7418017) or Delta (EPI_ISL_2378732) isolates at 37 °C for 1 h. The cells were next washed with DMEM and cultured in DMEM supplemented with 10% FCS and 1% penicillin–streptomycin and the serially diluted remdesivir (Selleck, S8932) or β-d-N4-hydroxycytidine (a derivative of molnupiravir; Cell Signalling Technology, 81178S). At 24 h after infection, the culture supernatants were collected and viral RNA was quantified using RT–qPCR. The assay of each compound was performed in quadruplicate, and the 50% neutralization titre was calculated using Prism 9 (GraphPad).

## Cytotoxicity assay

The CPE of the remdesivir and β-d-N4-hydroxycytidine was tested using the cell counting kit-8 (Dojindo, CK04-11) according to the manufacturer's instructions. One day before the assay, HOS-ACE2-TMPRSS2 cells (10,000 cells) were seeded into a 96-well plate. The cells were cultured with the serially diluted compound for 24 h. The cell counting kit-8 solution (10 µl) was added to each well, and the cells were incubated at 37 °C for 90 min. Absorbance was measured at 450 nm using the GloMax Explorer Microplate Reader (Promega).

## PV SG-PERT and infectivity

The Delta and Omicron spike-PVs were prepared as described in the 'Neutralization titre analyses' section and standardized using a SYBR Green-based product-enhanced PCR assay (SG-PERT) as described previously.[54] In brief, tenfold dilutions of virus supernatant were lysed at a 1:1 ratio in a 2× lysis solution (made up of 40% glycerol (v/v) 0.25% Triton X-100 (v/v) 100 mM KCl, RNase inhibitor 0.8 U ml$^{-1}$, TrisHCL 100 mM, buffered to pH 7.4) for 10 min at room temperature. Sample lysates (12 µl) were added to 13 µl of SYBR Green master mix (containing 0.5 µM of MS2-RNA Fwd and Rev primers, 3.5 pmol ml$^{-1}$ of MS2-RNA and 0.125 U µl$^{-1}$ of Ribolock RNase inhibitor) and cycled in a QuantStudio system. Relative amounts of reverse transcriptase activity were determined as the rate of transcription of bacteriophage MS2 RNA, with absolute RT activity calculated by comparing the relative amounts of RT to an RT standard of known activity.

PVs were used to transduce different target cell lines and 3D organoid cultures. After 48 or 72 h after transduction, the cells were lysed with direct addition of Bright Glo (Promega). The raw readings were then normalized to the SG-PERT and plotted using GraphPad Prism.

## Entry inhibitors in A549-ACE2-TMPRSS2 cells with PV and live virus

A549-ACE2/TMPRSS2 (A549-A2T2) cells were treated with either E64d (Tocris) or camostat (Sigma-Aldrich) for 2 h at each drug concentration before the addition of a comparable amount of input viruses pseudotyped with Delta or Omicron (approximately 100,000 RLU). The cells were then left for 48 h before addition of luciferase substrate (Promega) and read on a Glomax plate reader (Promega).

Quantification of viral replication in A549-ACE2/TMPRSS2-based luminescent reporter cells was performed as previously described (ref. [55]). In brief, A549-A2T2 reporter cells (clone E8) over-expressing *Renilla* luciferase (Rluc) and SARS-CoV-2 Papain-like protease-activatable circularly permuted firefly luciferase (FFluc) were seeded in flat-bottomed 96-well plates. The next day, the cells were treated with the indicated doses of camostat or E64d for 2 h, then infected with the Delta (EPI_ISL_1731019) or Omicron (M21021166) variant of SARS-CoV-2 at an m.o.i. of 0.01. After 24 h, the cells were lysed in Dual-Glo Luciferase Buffer (Promega) diluted 1:1 with PBS and 1% NP-40. Lysates were then transferred to opaque 96-well plates, and viral replication was quantified as the ratio of FFluc/Rluc luminescence measured using the Dual-Glo kit (Promega) according to the manufacturer's instructions. Percentage inhibition was calculated relative to the maximum luminescence signal for each condition, then analysed using the Sigmoidal, 4PL, X is log(concentration) function in GraphPad Prism.

## Immunofluorescence assays

HeLa-ACE2 cells were grown on 19 mm glass coverslips in 24-well plates and infected with Delta and Omicron for 24 h. Cells were fixed in 4% paraformaldehyde (Applichem), and permeabilized in PBS-T (0.1% (v/v) Triton X-100 in PBS) for 5 min. Cells were blocked in PBS-T containing 5% (w/v) BSA for 30 min and labelled with primary antibodies against SARS-CoV-2 Spike (GTX632604) or anti-GM130[EP892Y]-cis-Golgi (ab52649) in PBS-T and 1% (w/v) BSA for 1 h at room temperature. After washing, the cells were stained with fluorochrome-conjugated secondary antibodies in PBS-T, 1% (w/v) BSA for 1 h at room temperature in the dark. Nuclei were counterstained with DAPI and phalloidin 647 (A22287, Molecular Probes). Confocal images were acquired on a Leica TCS SP8 microscope. Post-acquisition analysis was performed using the Leica LAS X or Fiji (v.1.49) software. For distance analysis, the two-dimensional coordinates of the centroids of spike proteins were calculated using the Analyze Particles module of Fiji (ImageJ). The distance of each particle to the edge of the nucleus, visualized using DAPI staining, was looked up using a Euclidean distance map computed with the Distance Transform module of Fiji and exported as a list of distance measurements using the Analyze Particle function. Colocalization of Spike with the Golgi was estimated through Pearson's correlation coefficients ($R$) using the PSC colocalization plug-in (ImageJ, NIH). $R$ values between −1 (perfect negative correlation) and +1 (perfect positive correlation), with 0 corresponding to no correlation, were assigned[56]. Colocalization analyses were performed on >20 cells.

## Cell–cell fusion assay

Cell–cell fusion assays were carried out as follows. In brief, HEK293T GFP11 and Vero-GFP1-10 cells were seeded at 80% confluence at a 1:1 ratio in 48-well plates the day before. Cells were co-transfected with 0.5 µg of spike expression plasmids using Fugene 6 according to the manufacturer's instructions (Promega). Cell–cell fusion was measured using an Incucyte and determined as the proportion of green area to total phase area. Graphs were generated using Prism 8. To measure cell surface spike expression, HEK293 cells were transfected with *S* expression plasmids and stained with rabbit anti-SARS-CoV-2 S S1/S2 polyclonal antibodies (Thermo Fisher Scientific, PA5-112048, 1:100). Normal rabbit

IgG (SouthernBiotech, 0111-01, 1:100) was used as a negative control, and APC-conjugated goat anti-rabbit IgG polyclonal antibodies (Jackson ImmunoResearch, 111-136-144, 1:50) were used as secondary antibodies. The surface expression level of S proteins was analysed using FACS Canto II (BD Biosciences) and FlowJo v.10.7.1 (BD Biosciences). The gating strategy for flow cytometry is shown in Supplementary Fig. 8.

### Analysis of the size of infection foci

Vero E6 cells (ATCC CRL-1586, obtained from Cellonex) were propagated in complete growth medium consisting of DMEM with 10% fetal bovine serum (Hyclone) containing 10 mM HEPES, 1 mM sodium pyruvate, 2 mM L-glutamine and 0.1 mM nonessential amino acids (Sigma-Aldrich). Vero E6 cells were passaged every 3–4 days. H1299 cell lines were propagated in growth medium consisting of complete RPMI 1640 medium with 10% fetal bovine serum containing 10 mM HEPES, 1 mM sodium pyruvate, 2 mM L-glutamine and 0.1 mM non-essential amino acids. H1299 cells were passaged every second day. The H1299-E3 (H1299-ACE2, clone E3) cell line was derived from H1299 (CRL-5803).

Virus stock using a South Africa-derived (EPI_ISL_7886688) isolate, and Delta variant isolate, was produced as described previously[2] and used at approximately 50–100 focus-forming units per microwell. H1299-E3 cells were plated in a 96-well plate (Corning) at 30,000 cells per well 1 day before infection. Cells were infected with 100 μl of the virus working stock for 1 h, then 100 μl of a 1× RPMI 1640 (Sigma-Aldrich, R6504), 1.5% carboxymethylcellulose (Sigma-Aldrich, C4888) overlay was added without removing the inoculum. Cells were fixed 2 h or 18 h after infection using 4% PFA (Sigma-Aldrich) for 20 min. Foci were stained with a rabbit anti-spike monoclonal antibody (BS-R2B12, GenScript A02058) at 0.5 μg ml⁻¹ in a permeabilization buffer containing 0.1% saponin (Sigma-Aldrich), 0.1% BSA (Sigma-Aldrich) and 0.05% Tween-20 (Sigma-Aldrich) in PBS. Plates were incubated with primary antibodies overnight at 4 °C, then washed with wash buffer containing 0.05% Tween-20 in PBS. Secondary goat anti-rabbit HRP-conjugated antibodies (Abcam, ab205718) was added at 1 μg ml⁻¹ and incubated for 2 h at room temperature with shaking. TrueBlue peroxidase substrate (SeraCare, 5510-0030) was then added at 50 μl per well and incubated for 20 min at room temperature. Plates were imaged using the ImmunoSpot Ultra-V S6-02-6140 Analyzer ELISPOT instrument with BioSpot Professional built-in image analysis (CTL). For microscopy, wells were imaged using a Metamorph-controlled Nikon TiE motorized microscope with a ×10 objective. Automated image analysis was performed using a custom script in MATLAB 2019b. Statistics and graphing were performed using Prism v.8.0.

### Reporting summary

Further information on research design is available in the Nature Research Reporting Summary linked to this paper.

### Data availability

The sequences of the virus isolates used are available at GISAID: B.1.1.617.2 (Delta, GISAID: EPI_ISL_1731019), B.1.1.529 (Omicron, BA.1 isolate from the UK; ENA project no PRJEB50520, accession number ERS10754850), B.1.1.529 (Omicron, BA.1 isolate from South Africa, GISAID: EPI_ISL_7886688) and (BA.1 isolate from Japan, GISAID: EPI_ISL_7418017). The structural models used in this study for Delta and Omicron variants are available at GitHub (https://github.com/CSB-Thukral-Lab/Spike_structural_models_Delta_and_Omicron and https://github.com/ojcharles/viral_alphafold). The scRNA-seq data analysis is available at GitHub (https://github.com/elo073/Lung5loc_scRNAseq_omcr_entrygenes). Source data are provided with this paper.

### Code availability

The code for scRNA-seq data analysis is available at GitHub (https://github.com/elo073/Lung5loc_scRNAseq_omcr_entrygenes).

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

**Acknowledgements** We thank P. Lehner for Calu-3 cells; J. Voss for HeLa ACE2; S. Cook for H1299 and S. Rihn for the A549-ACE2/TMPRSS2 cells; G. Screaton for the Omicron isolate M21021166; C. Cormie for microscope training; A. Albecka for advice on SARS-CoV2 virion purification; K. Khan for work on foci; T. de Silva for the Delta variant isolate; all of the members belonging to The Genotype to Phenotype Japan (G2P-Japan) Consortium; the staff at the National Institute for Infectious Diseases, Japan and H. Arase (Osaka University, Japan) for providing virus isolates and reagents; and the staff at Ituro Inoue and Sachiko Sakamoto (National Institute of Genetics, Japan) for technical support. R.K.G. is supported by a Wellcome Trust Senior Fellowship in Clinical Science (WT108082AIA). I.G. is a Wellcome Senior Fellow (207498/Z/17/Z). This study was supported by the Cambridge NIHRB Biomedical Research Centre. I.A.T.M.F. is funded by a SANTHE award (DEL-15-006). A.A. is supported by a Africa Research Excellence Fund Research Development Fellowship (AREF-318-ABDUL-F-C0882). This work was supported by the MRC (TSF, MR/T032413/1 to N.J.M.), NHSBT (WPA15-02 to N.J.M.), Addenbrooke's Charitable Trust (900239 to N.J.M.). The authors acknowledge support from the G2P-UK National Virology consortium funded by MRC/UKRI (MR/W005611/1) and thank T. Peacock and W. Barclay. This study was also supported by The Rosetrees Trust and the Geno2pheno UK consortium. S.F. acknowledges the EPSRC (EP/V002910/1). K. Sato is supported by AMED Research Program on Emerging and Re-emerging Infectious Diseases (20fk0108270 and 20fk0108413), JST SICORP (JPMJSC20U1 and JPMJSC21U5) and JST CREST (JPMJCR20H4). This study was supported in part by AMED Research Program on Emerging and Re-emerging Infectious Diseases 20fk0108163 (to A. Saito), 20fk0108146 (to K. Sato), 20fk0108270 (to K. Sato), 20fk0108413 (to T. Ikeda and K. Sato) and 20fk0108451 (to A. Saito, T. Ikeda, T.U., A.T.-K., the G2P-Japan Consortium and K. Sato); the AMED Research Program on HIV/AIDS 21fk0410033 (to A. Saito) and 21fk0410039 (to K. Sato); the AMED Japan Program for Infectious Diseases Research and Infrastructure 20wm0325009 (to A. Saito) and 21wm0325009 (to A. Saito); JST A-STEP JPMJTM20SL (to T. Ikeda); JST SICORP (e-ASIA) JPMJSC20U1 (to K. Sato); JST SICORP JPMJSC21U5 (to K. Sato), JST CREST JPMJCR20H4 (to K. Sato); JSPS KAKENHI Grant-in-Aid for Scientific Research C 19K06382 (to A. Saito), Scientific Research B 18H02662 (to K. Sato) and 21H02737 (to K. Sato); JSPS Fund for the Promotion of Joint International Research (Fostering Joint International Research) 18KK0447 (to K. Sato); the JSPS Core-to-Core Program JPJSCCA20190008 (A. Advanced Research Networks) (to K. Sato); JSPS Research Fellow DC1 19J20488 (to I.K.); the JSPS Leading Initiative for Excellent Young Researchers (LEADER) (to T. Ikeda); the Takeda Science Foundation (to T. Ikeda); the Tokyo Biochemical Research Foundation (to K. Sato); the Mitsubishi Foundation (to T. Ikeda); the Shin-Nihon Foundation of Advanced Medical Research (to T. Ikeda); a Grant for Joint Research Projects of the Research Institute for Microbial Diseases, Osaka University (to A. Saito); an intramural grant from Kumamoto University COVID-19 Research Projects (AMABIE) (to T. Ikeda); the Intercontinental Research and Educational Platform Aiming for Eradication of HIV/AIDS (toT. Ikeda); and the Joint Usage/Research Center program of Institute for Frontier Life and Medical Sciences, Kyoto University (to K. Sato). This study was supported by the National Institute of Allergy and Infectious Diseases (DP1AI158186 and HHSN272201700059C to D.V.), a Pew Biomedical Scholars Award (to D.V.), an Investigators in the Pathogenesis of Infectious Disease Awards from the Burroughs Wellcome Fund (to D.V.), Fast Grants (to D.V.) and the Bill and Melinda Gates Foundation (OPP1156262 to D.V.). D.V. is an Investigator of the Howard Hughes Medical Institute. T.B. is supported by an EASL Juan Rodès fellowship. F.S. was supported by a UKRI Future Leaders fellowship. A. Sigal. was supported by the Bill and Melinda Gates award INV-018944, National Institutes of Health award R01 AI138546 and a South African Medical Research Council award 6084-CO-AP-2020. S.K.Z. was supported by the National Institute of General Medical Sciences (5T32GM008268-32). Studies using human donor lung tissue were supported by the NIHR Blood and Transplant Research Unit (BTRU) in Organ

Donation and Transplantation at Newcastle University and the University of Cambridge and in partnership with NHS Blood and Transplant (NHSBT). The views expressed are those of the author(s) and not necessarily those of the NIHR, the Department of Health and Social Care or NHSBT. We thank NIHR BioResource volunteers for their participation, and gratefully acknowledge NIHR BioResource centres, NHS Trusts and staff for their contribution. We thank the National Institute for Health Research, NHS Blood and Transplant, and Health Data Research UK as part of the Digital Innovation Hub Programme. The work was partly funded by awards from NIHR to the NIHR BioResource (RG94028 & RG85445)

**Author contributions** Conceived research and designed study: R.K.G., K. Sato, B.M., N.J.M. and I.G.G. Designed experiments: D.C., M.S.P., A.d.M., B.M., R.K.G., N.G., M.H., I.G.G., J.H.L., A.T.-K., K. Sato, T.U., S.M., J.K., T.S.T., M.T., S.K.Z., A.C.W., D.V., F.S., A. Sigal, L.C.J., A.J.F., S.A.T., R.D., J.B., J.A.G.B. and K.G.C.S. Performed experiments: B.M., A.A., I.A.T.M.F., N.G., S.K.Z., A.d.M., S.A.K., S.F., S.R., P.P.G., D.Y., I.K., A. Saito, G.P., O.J.C., D.L.M., T.B., A.T.-K., K. Shirakawa, T.U., S.M., J.K., T.S.T., M.T., T. Ikeda, J.E.B., A.J., L.C.-G., P.M. and E.M. Analysed data: S.K.Z., N.G., A.C.W., D.V., R.K.G., B.M., I.G.G., N.J.M., P.P.G., D.Y., I.K., A. Saito, D.V., A.T.-K., K. Shirakawa, T.U., S.M., J.K., T.S.T., M.T., A. Sigal, K.M. and M.S.P.; R.K.G., D.V. and B.M. wrote the manuscript with input from all of the authors.

**Competing interests** M.S.P., D.C. and A.d.M. are employees of Vir Biotechnology and may hold shares in Vir Biotechnology. R.K.G. has received honoraria for educational activities from Johnson & Johnson, ViiV and GSK. F.S. is shareholder in Bilitech. The Veesler laboratory has received an unrelated research sponsored agreement from Vir Biotechnology. KGCS is a member of the GSK Immunology Scientific Advisory Board, is a founder, Chief Medical Officer, and a non-executive director of PredictImmune, and is a co-founder of Rheos Medicines. The other authors declare no competing interests.

**Additional information**
**Correspondence and requests for materials** should be addressed to Kei Sato or Ravindra K. Gupta.

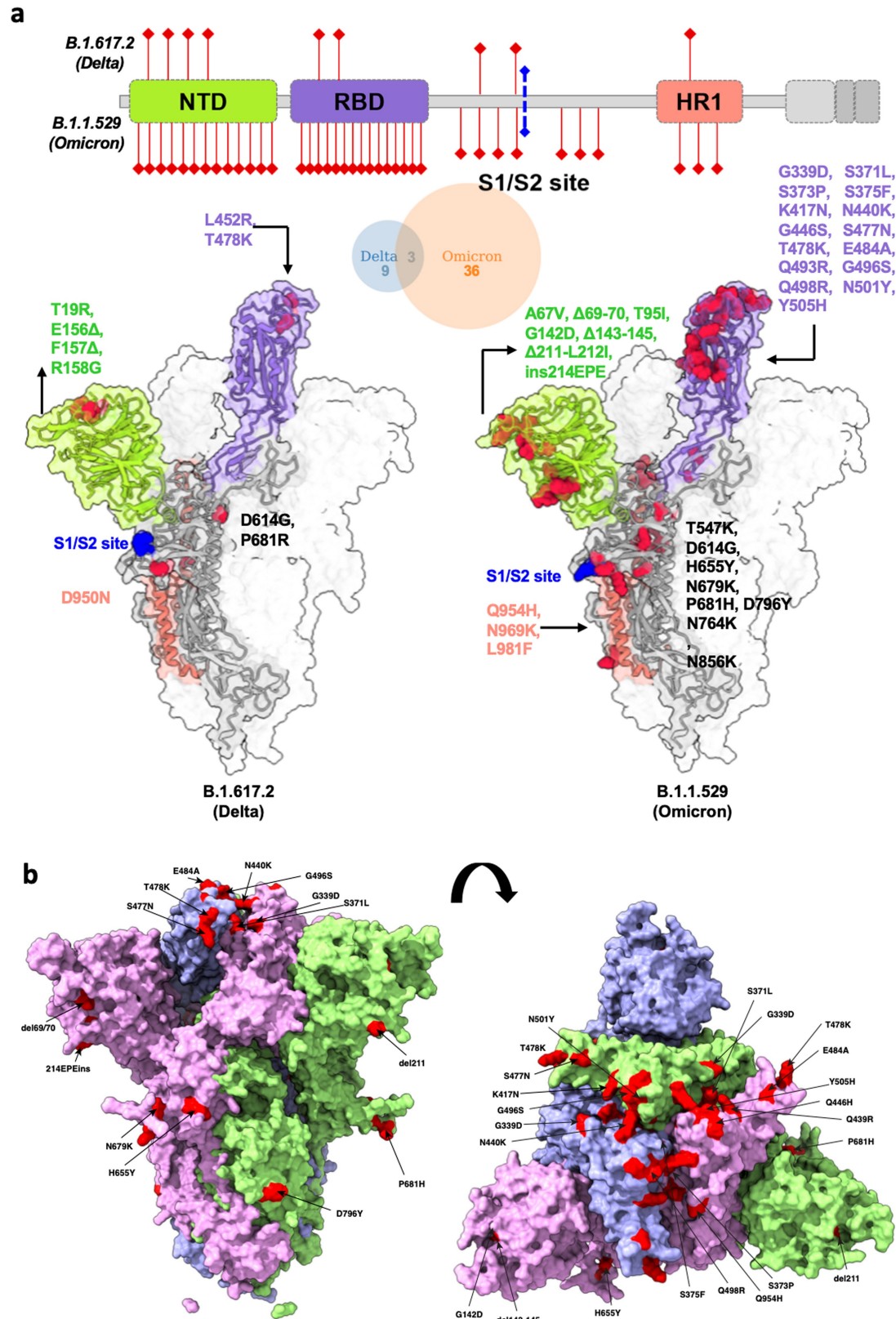

**Extended Data Fig. 1 | Structural model of SARS-CoV-2 Delta and Omicron spike variants. a**. Variant structures for Delta and Omicron of spike protein displaying mutational sites in red colour, generated using the cryo-EM structure PDB# 7A94 as a reference. The schematic highlights differences in domain-wise mutations across protein length. **b**. Side and top down surface representation of the Omicron spike protein. Spike homotrimer structures were created predicted *in silico* by the Alphafold2 software package. Individual mutations making up the Omicron spike are highlighted in red on each of the three homotrimers.

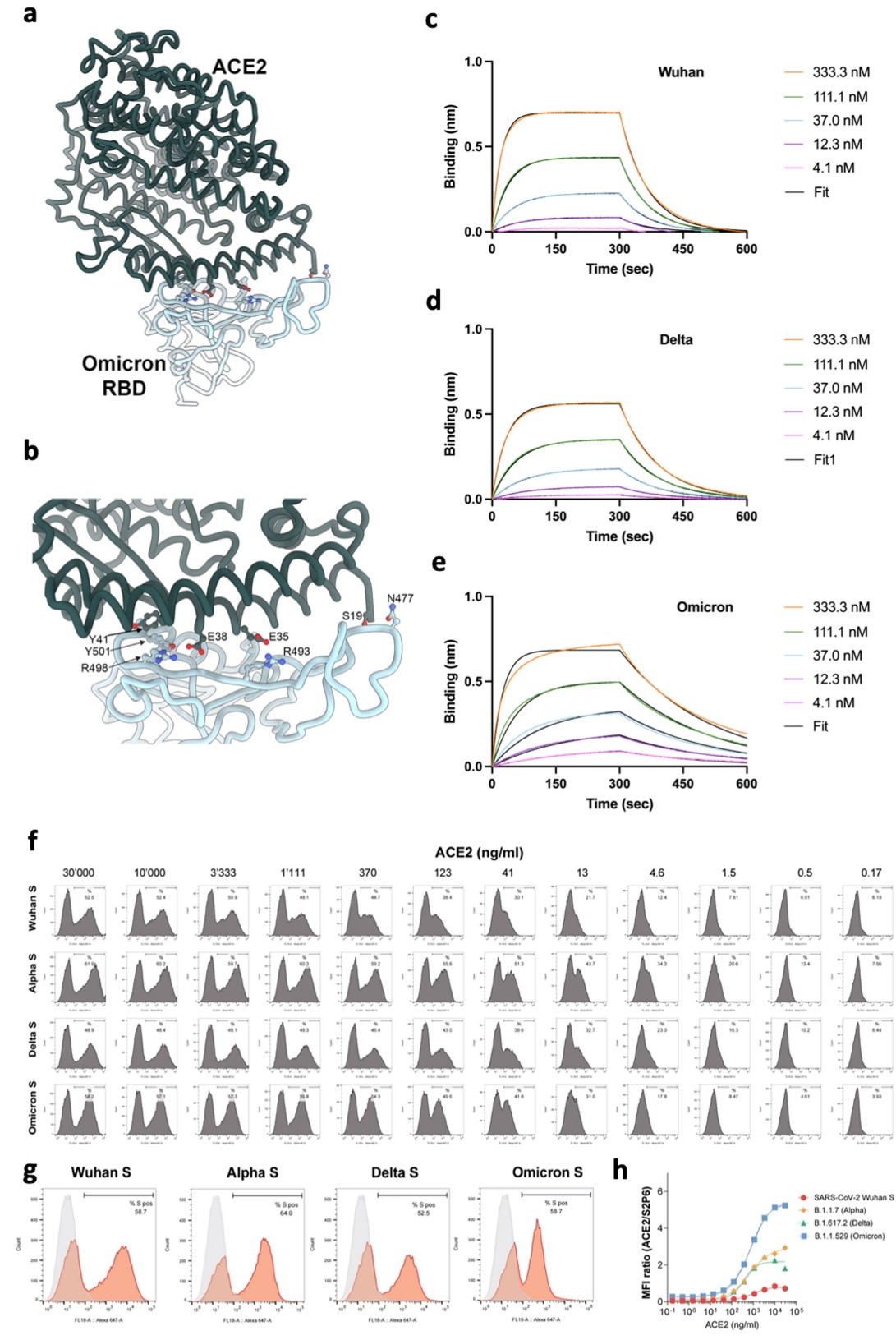

**Extended Data Fig. 2** | See next page for caption.

**Extended Data Fig. 2 | Omicron spike binding to ACE2. a**. Rendering of the contacts formed between the Omicron RBD (light blue) and human ACE2 (dark green) using PDB 7TN0 described in[15] **b**. Zoomed-in view showing select amino acid side chains participating in remodelling of interactions between the Omicron RBD and ACE2. **c**–**e**. Biolayer interferometry binding analysis of human ACE2 to immobilized SARS-CoV-2 Wuhan-Hu-1 **c**. Delta **d**. or Omicron **e**. RBDs. Black lines correspond to a global fit of the data using a 1:1 binding model. **f**. Flow cytometry analysis of ACE2 binding to SARS-CoV-2 Wuhan, B.1.1.7 (Alpha), B.1.617.2 (Delta) or Omicron (B.1.1.529) spike proteins transiently expressed in ExpiCHO cells. **g**. Expression level of each spike as measured via binding by S2-subunit targeting S2P6 mAb. Grey histograms represent background florescence of cells stained with the secondary antibody only. **h**. Normalized ACE2 binding results based on spike protein expression levels. Values on the Y axis indicate the ratio between the mean fluorescence intensity (MFI) of positive cells stained for ACE2 binding and the MFI of the S2P6 positive cells. Data points in h. are mean of technical triplicates and data shown are representative two independent experiments.

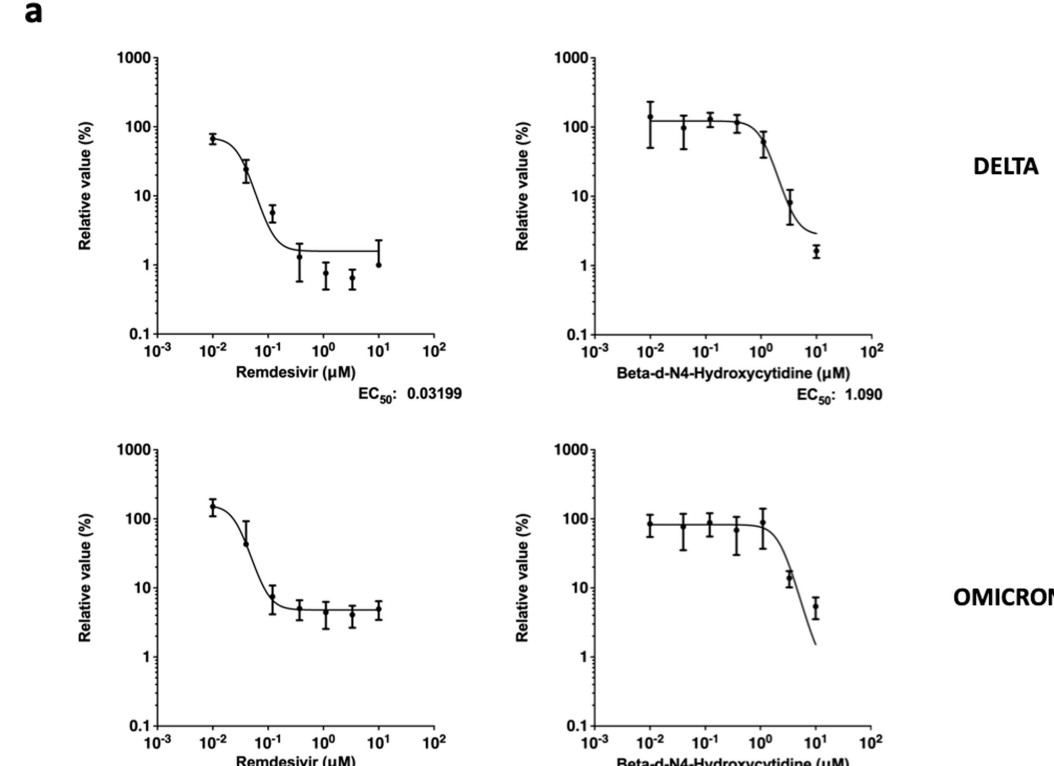

**DELTA**

**OMICRON**

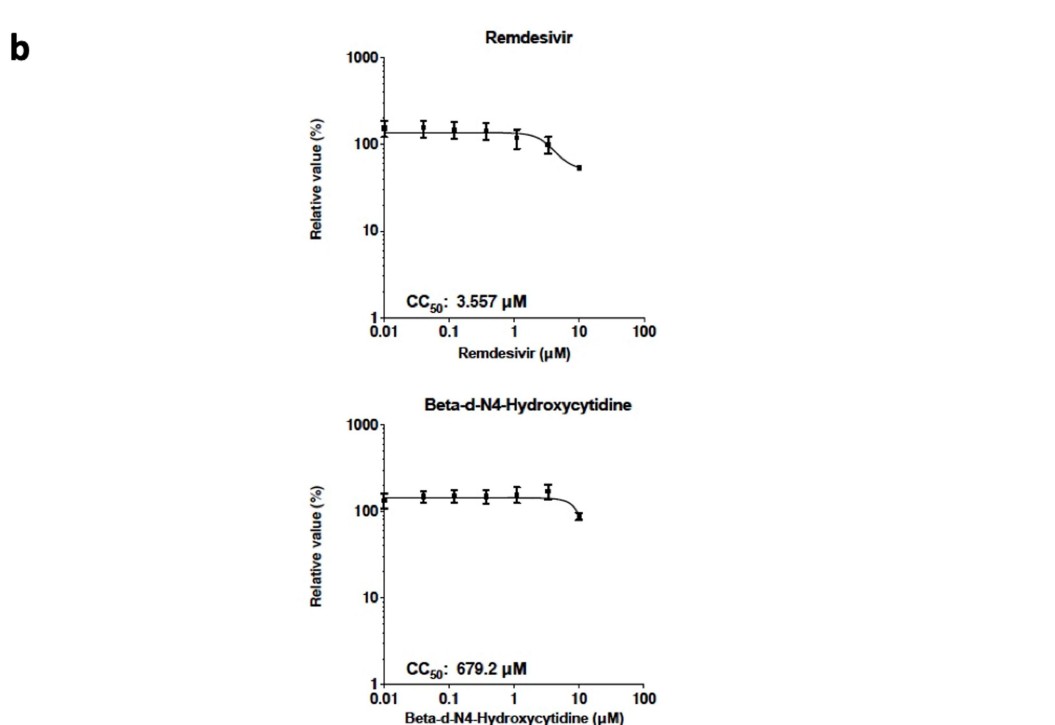

**Extended Data Fig. 3 | Sensitivity of replication competent SARS-CoV-2 Omicron and Delta variants to clinically approved direct acting antiviral molecules remdesivir and the active metabolite of molnupiravir.** HOS cells overexpressing ACE2 and TMPRSS2 were used and a viral input of 1,000 $TCID_{50}$ was used ($TCID_{50}$ measured using VeroE6/TMPRSS2 cells). **a**. Dose-response curves. Infection as measured by viral RNA copies relative to the no drug control (100%) is plotted on the y axis with serial drug dilution on the x axis.

EC50 is indicated for each panel and calculated in GraphPad Prism. Data points are mean of technical quadruplicates with +/- SEM shown. Curve fitting for dose response was done in GraphPad prism. **b**. toxicity assay showing relative cell viability at a range of drug doses. Data points are mean of technical quadruplicates with +/- SEM shown. All data are representative of two independent experiments.

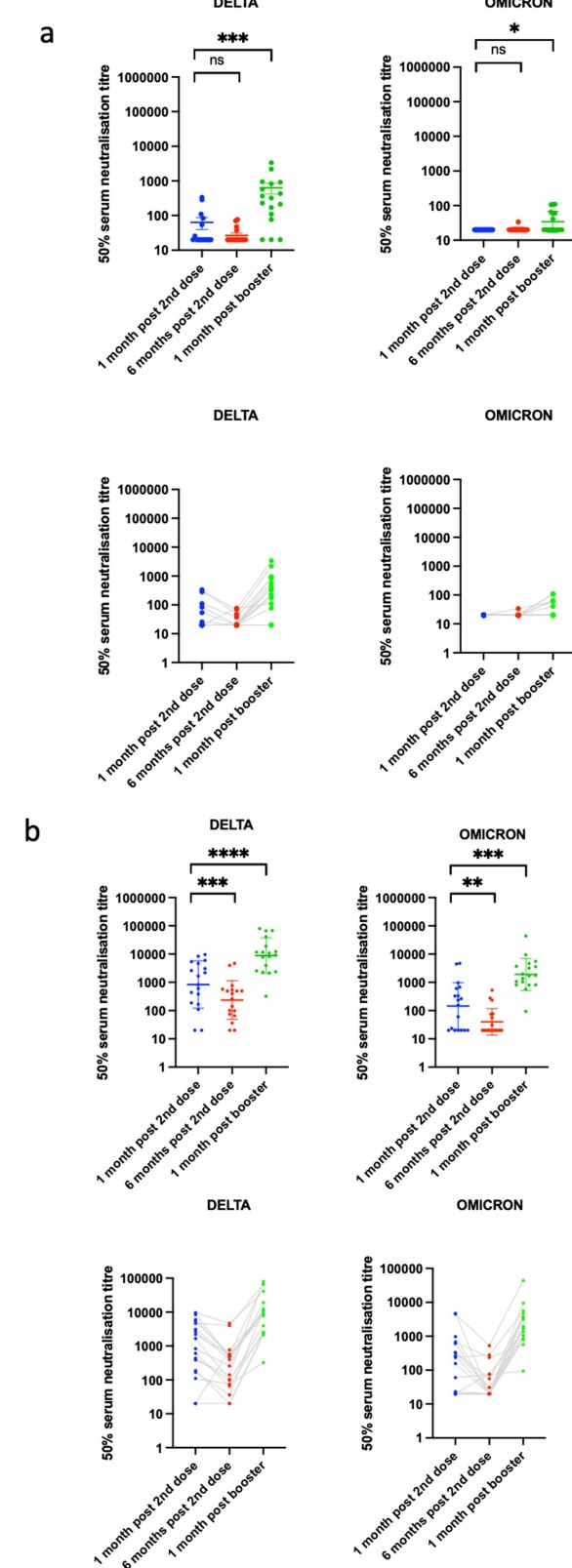

**Extended Data Fig. 4 | Neutralization of spike pseudotyped virus by sera from vaccinated individuals over three time points following dose two (ChAdOx-1 or BNT162b2) and dose three (BNT162b2 only). a.** n = 17 ChAdOx-1 or **b.** n = 18 BNT12b2. GMT (geometric mean titre) shown as a line with s.d as bars are presented. Data representative of two independent experiments each with two technical replicates. **p < 0.01, *** p < 0.001, ****p < 0.0001 Wilcoxon matched-pairs signed rank test, ns not significant, two-sided. Data are for individuals who tested negative for anti-N IgG at each time point.

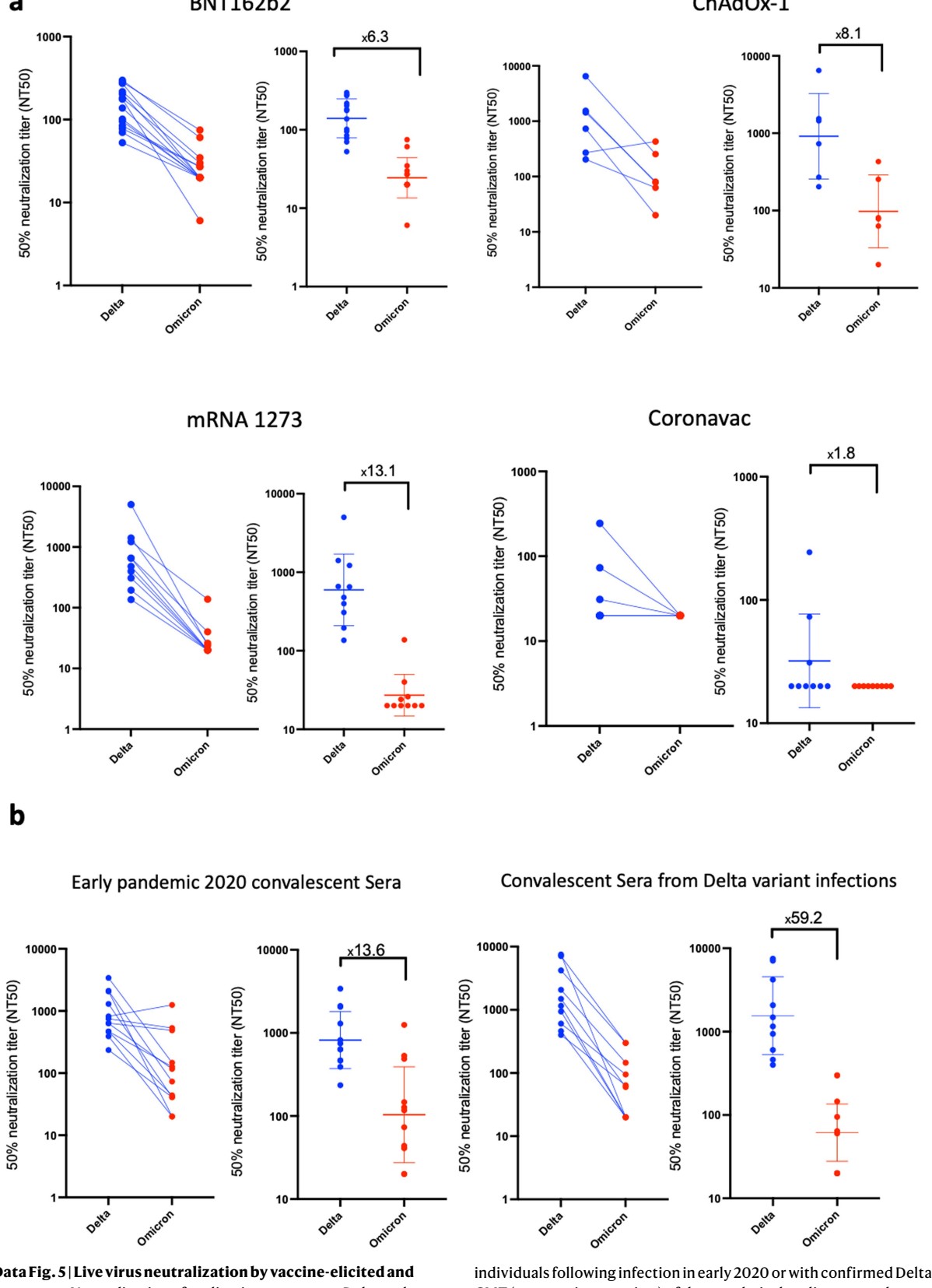

**Extended Data Fig. 5 | Live virus neutralization by vaccine-elicited and convalescent sera. a**. Neutralization of replication competent Delta and Omicron viruses by sera from vaccinated individuals following dose two of ChAdOx-1 (n = 5), BNT162b2 (n = 10), mRNA 1273 (n = 10), and Coronavac (n = 9) vaccines. **b**. neutralization of live viruses by sera derived from recovered individuals following infection in early 2020 or with confirmed Delta infection. GMT (geometric mean titre) of three technical replicates are shown as a line with s.d as bars are presented. Data representative of two independent experiments. Fold change in NT50 is indicated.

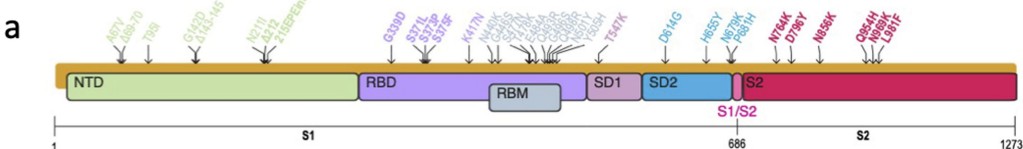

**Extended Data Fig. 6 | SARS-CoV-2 Omicron Variant spike pseudotyped lentiviruses. a**. Graphical representation of Omicron spike mutations present in expression plasmid used to generate lentiviral pseudotyped virus (PV). Mutations coloured according to location in spike; bold mutations are novel to this lineage and have not been identified in previous VOCs. **b**. western blots of pseudotyped virus (PV) virions from 293T producer cells following transfection with plasmids expressing lentiviral vectors and SARS-CoV-2 S plasmids. (WT- Wuhan-1 with D614G), probed with antibodies for HIV-1 p24 and

SARS-Cov-2 S2 (top) and S1 (bottom). **c**, **d**. quantification of western blots showing **c**. ratio of spike:p24 in virions, **d**. ratio of S2:total spike. Mean ratio +/- sd is shown for 3 biological replicates **e**. Western blot of cell lysates used to produce virions with **f**. quantification of S2 to total spike ratio. Mean ratio +/- sd is shown for 3 biological replicates **g**. brightfield images of lower airway organoids **h**. brightfield images of of cholangiocyte organoids (Scale bars 200 μm). **i**. infection schematic for entry assays in organoids created with BioRender.com.

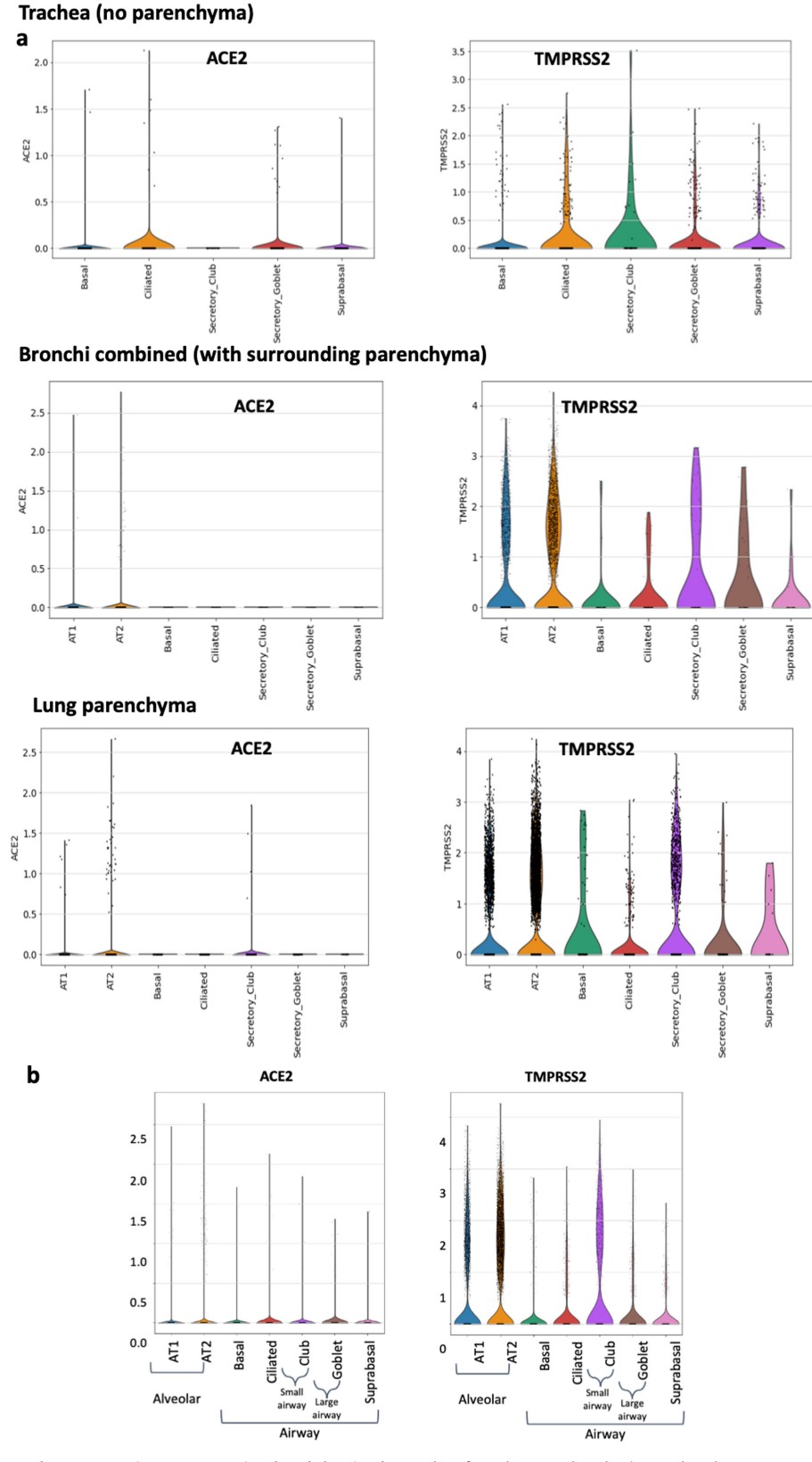

**Extended Data Fig. 7 | ACE2 and TMPRSS2 tissue expression levels by single cell RNAseq. a.** Log-normalized expression of ACE2 and TMPRSS2 genes in single-nucleus RNAseq data for indicated tissue types and. **b.** Log-normalized aggrerated expression of ACE2 and TMPRSS2 genes in single-nucleus RNAseq data from human alveolar (AT1 - alveolar type 1, AT2 - alveolar type 2 pneumocytes), and airway epithelial cells (basal, suprabasal, goblet and ciliated). Data are derived from in Madissoon et al.

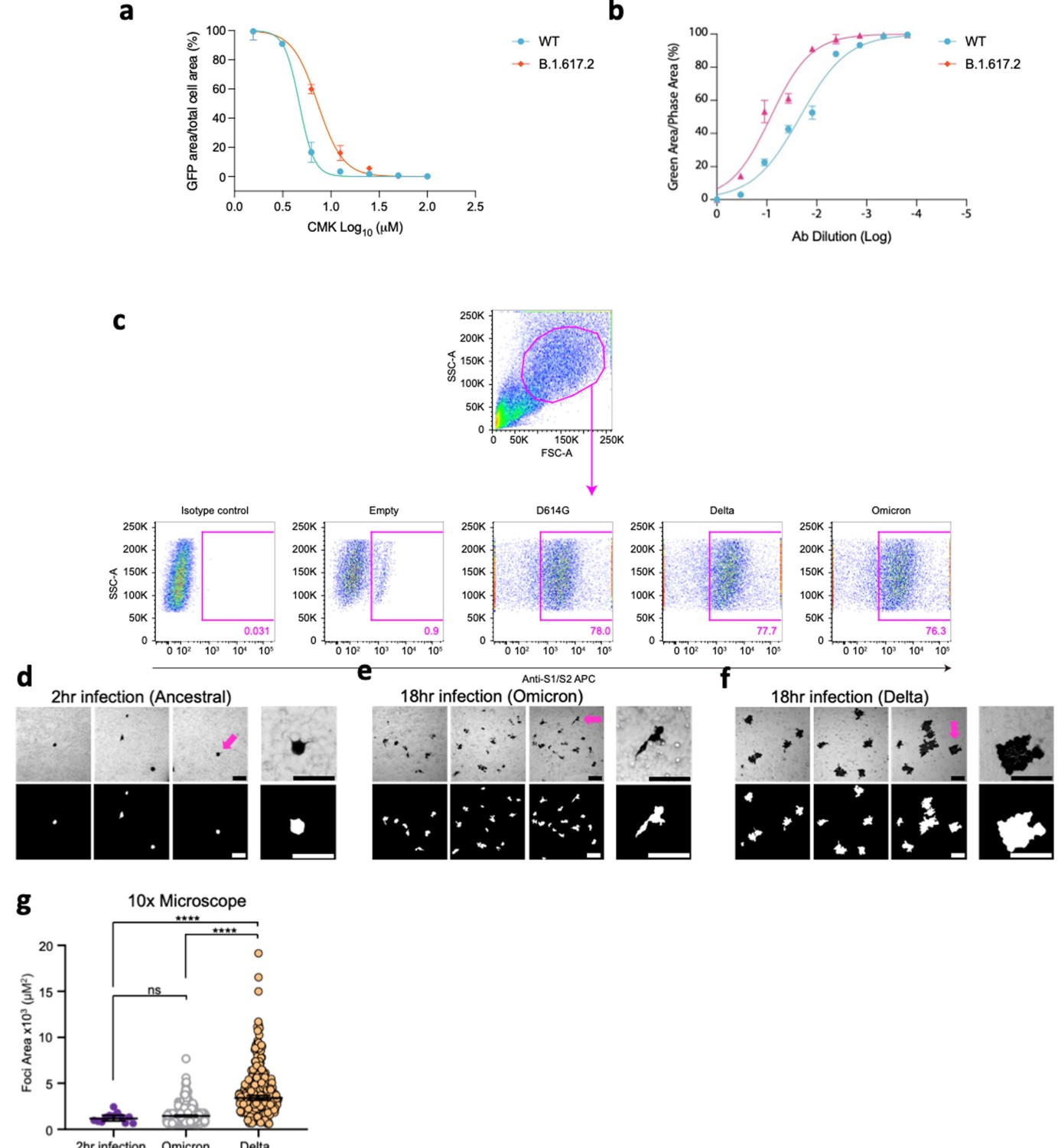

**Extended Data Fig. 8 | SARS-CoV-2 cell-cell fusion and infection focus formation. a.** effect of furin inhibition using CMK on cell-cell fusion for WT (Wuhan-1 D614G) and Delta variant. CMK drug dilution indicated on x axis and fusion on y axis. Quantification of cell-cell fusion shows percentage of green area to total cell area. Mean of technical replicates is plotted with error bars representing SEM. Data are representative of at least two independent experiments. **b.** impact of neutralising antibodies from UK wave one 2020 sera on cell-cell fusion for WT (Wuhan-1 D614G) and Delta variant. Serial dilutions of sera added to acceptor cells before co-cultivation with donor cells. Data points are mean of technical replicates is plotted with error bars representing SEM. **c.** Gating strategy for cell surface staining of SARS-CoV-2 spike proteins from

293T cells transfected with plasmids expressing spike, using rabbit anti-SARS-CoV-2 spike S1/2 polyclonal antibody at 1:100. **d–g.** Omicron shows attenuated localized spread compared to Delta. **d.** Microscope images of foci (pink arrow) 2 h post-infection with ancestral SARS-CoV-2. Bar is 200 microns. **e.** Images of Omicron foci 18 h post-infection. **f.** Images of Delta foci 18 h post-infection **g.** Geometric mean and 95% confidence intervals of focus area of microscope images of 2 h post-ancestral infection (n = 12 foci, purple), 18 h post-Omicron infection (n = 363, grey), and 18 h post-Delta infection (n = 275, orange). p-values are not significant (ns) or **** < 0.0001 as determined by two sided Wilcoxon rank sum test.

**Extended Data Table 1 | Kinetic Analysis of human ACE2 binding to SARS-CoV-2 VOC by Biolayer Interferometry**

|  | Wuhan | Delta | Omicron |
|---|---|---|---|
| **KD (nM)** | 127 | 190 | 44 |
| **$k_{on}$ ($M^{-1}$ $s^{-1}$)** | $1.1 \times 10^5$ | $5.9 \times 10^4$ | $1.1 \times 10^5$ |
| **$k_{off}$ ($s^{-1}$)** | $1.4 \times 10^{-2}$ | $1.1 \times 10^{-2}$ | $4.7 \times 10^{-3}$ |

Values reported represent the global fit to the data shown in Extended Data Fig. 2.

**Extended Data Table 2 | Demographic characteristics of participants in longitudinal vaccine-elicited sera neutralization study**

| Characteristic | Total | Pfizer (BNT162b2) | AstraZeneca (ChAdOx1) |
|---|---|---|---|
| Total number, n (%) | 40 | 20 (50) | 20 (50) |
| Age, median years (IQR) | 70.5 (62.5, 74.0) | 71.0 (55.8, 75.8) | 69.5 (63.0, 73.8) |
| ≥80, n (%) | 4 (10) | 4 (20) | 0 (0) |
| <80, n (%) | 36 (90) | 16 (80) | 20 (100) |
| Gender, n (%) | | | |
| Male | 15 (37.5) | 8 (40) | 7 (35) |
| Female | 25 (62.5) | 12 (60) | 13 (65) |
| Previous COVID-19 | | | |
| Yes | 5 (12.5) | 2 (10) | 3 (15) |
| No | 35 (87.5) | 18 (90) | 17 (85) |
| Type of booster | | | |
| Moderna | 3 (7.5) | 0 | 3 (15) |
| Pfizer | 37 (92.5) | 20 (100) | 17 (85) |
| Duration between booster dose and sampling, median days (IQR) | 34 (32, 36) | 34.0 (32.0, 36.0) | 33.5 (30.3, 35.8) |

Data relate to participants in Fig. 1 and Extended Data Fig. 4. Categorical variables (expressed as proportions and percentages) and continuous variables (expressed as medians with inter quartile ranges [IQR]) are shown.

# Reporting Summary

## Statistics

For all statistical analyses, confirm that the following items are present in the figure legend, table legend, main text, or Methods section.

| n/a | Confirmed | |
|---|---|---|
| ☐ | ☒ | The exact sample size (*n*) for each experimental group/condition, given as a discrete number and unit of measurement |
| ☐ | ☒ | A statement on whether measurements were taken from distinct samples or whether the same sample was measured repeatedly |
| ☐ | ☒ | The statistical test(s) used AND whether they are one- or two-sided<br>*Only common tests should be described solely by name; describe more complex techniques in the Methods section.* |
| ☒ | ☐ | A description of all covariates tested |
| ☒ | ☐ | A description of any assumptions or corrections, such as tests of normality and adjustment for multiple comparisons |
| ☐ | ☒ | A full description of the statistical parameters including central tendency (e.g. means) or other basic estimates (e.g. regression coefficient) AND variation (e.g. standard deviation) or associated estimates of uncertainty (e.g. confidence intervals) |
| ☐ | ☒ | For null hypothesis testing, the test statistic (e.g. *F*, *t*, *r*) with confidence intervals, effect sizes, degrees of freedom and *P* value noted<br>*Give P values as exact values whenever suitable.* |
| ☒ | ☐ | For Bayesian analysis, information on the choice of priors and Markov chain Monte Carlo settings |
| ☒ | ☐ | For hierarchical and complex designs, identification of the appropriate level for tests and full reporting of outcomes |
| ☐ | ☒ | Estimates of effect sizes (e.g. Cohen's *d*, Pearson's *r*), indicating how they were calculated |

*Our web collection on statistics for biologists contains articles on many of the points above.*

## Software and code

Policy information about availability of computer code

| Data collection | *Provide a description of all commercial, open source and custom code used to collect the data in this study, specifying the version used OR state that no software was used.* |
|---|---|
| Data analysis | Prism 9, Excel, GROMACS2018; FloJo, Fiji (v1.49) software, ImageJ, geneious prime, scanpy, Alphafold, GISAID and EpiCoV websites |

For manuscripts utilizing custom algorithms or software that are central to the research but not yet described in published literature, software must be made available to editors and reviewers. We strongly encourage code deposition in a community repository (e.g. GitHub). See the Nature Portfolio guidelines for submitting code & software for further information.

## Data

Policy information about availability of data

All manuscripts must include a data availability statement. This statement should provide the following information, where applicable:
- Accession codes, unique identifiers, or web links for publicly available datasets
- A description of any restrictions on data availability
- For clinical datasets or third party data, please ensure that the statement adheres to our policy

The raw data in this study are available from the corresponding authors with no restrictions upon reasonable request.

# Field-specific reporting

Please select the one below that is the best fit for your research. If you are not sure, read the appropriate sections before making your selection.

☒ Life sciences ☐ Behavioural & social sciences ☐ Ecological, evolutionary & environmental sciences

For a reference copy of the document with all sections, see nature.com/documents/nr-reporting-summary-flat.pdf

# Life sciences study design

All studies must disclose on these points even when the disclosure is negative.

| | |
|---|---|
| Sample size | Our sample sizes were based on our previous in vitro SARS-CoV-2 work, and based on practical availability of sera from participants. |
| Data exclusions | Individuals positive for N antibody were excluded from analysis in neutralisation experiments and data with and without the exclusions are presented in the figures |
| Replication | experiments were done at least twice as biological replicates |
| Randomization | we did not use randomisation in our lab experimental designs |
| Blinding | we did not use blinding in our lab experimental designs |

# Reporting for specific materials, systems and methods

We require information from authors about some types of materials, experimental systems and methods used in many studies. Here, indicate whether each material, system or method listed is relevant to your study. If you are not sure if a list item applies to your research, read the appropriate section before selecting a response.

## Materials & experimental systems

| n/a | Involved in the study |
|---|---|
| ☐ | ☒ Antibodies |
| ☐ | ☒ Eukaryotic cell lines |
| ☒ | ☐ Palaeontology and archaeology |
| ☒ | ☐ Animals and other organisms |
| ☐ | ☒ Human research participants |
| ☒ | ☐ Clinical data |
| ☒ | ☐ Dual use research of concern |

## Methods

| n/a | Involved in the study |
|---|---|
| ☒ | ☐ ChIP-seq |
| ☐ | ☒ Flow cytometry |
| ☒ | ☐ MRI-based neuroimaging |

## Antibodies

| | |
|---|---|
| Antibodies used | For Western blot:<br>Rabbit anti-SARS-CoV-2 S polyclonal antibody (PA1-41165, Thermofisher, 1:2,000),<br>Rabbit anti-SARS-CoV-2 N monoclonal antibody (clone HL344, GeneTex, Cat# GTX635679, 1:2,000),<br>Mouse anti-SARS-CoV-2 S1 monoclonal antibody (MAB105403, R&D systems, 1:2,000),<br>Rabbit anti-GAPDH polyclonal antibody (10494-1-AP, Proteintech, 1:5,000),<br>Mouse anti-HIV p24 monoclonal antibody (NIBSC, ARP313, 1:20,000)<br>Horseradish peroxidase (HRP)-conjugated Goat anti-rabbit IgG antibody (Cell Signaling, 7074, 1:5,000),<br>HRP-conjugated anti-mouse IgG antibody (Cell Signaling, 7076, 1:5,000).<br><br>For neutralisation assay:<br>Casirivimab and Imdevimab<br><br>For IF<br>Mouse anti-SARS-CoV-2 Spike monoclonal antibody (clone1A9, GeneTex, GTX632604,)<br>anti-GM130[EP892Y]-cis-Golgi (ab52649, 1:250)<br>phalloidin 647 (Thermofisher, A22287, 1:1000)<br>Goat anti-Rabbit IgG (H+L) Cross-Adsorbed Secondary Antibody, Alexa Fluor 594 (Thermofisher, A-11012, 1:1,000)<br>Goat anti-Mouse IgG (H+L) Cross-Adsorbed Secondary Antibody, Alexa Fluor 488 (Thermofisher, A-11001, 1:1,000)<br><br>For FACS<br>Alexa Fluor647 Goat Anti-Human IgG secondary Ab (1.5 mg/ml) (Jackson Immunoresearch)<br>rabbit anti-SARS-CoV-2 S S1/S2 polyclonal antibody (Thermo Fisher Scientific, Cat# PA5-112048, 1:100).<br>Normal rabbit IgG (SouthernBiotech, Cat# 0111-01, 1:100)<br>APC-conjugated goat anti-rabbit IgG polyclonal antibody (Jackson ImmunoResearch, Cat# 111-136-144, 1:50) |

| Validation | Rabbit anti-SARS-CoV-2 S polyclonal antibody (PA1-41165, Thermofisher, 1:2,000),<br>Rabbit anti-SARS-CoV-2 N monoclonal antibody (clone HL344, GeneTex, Cat# GTX635679, 1:2,000),<br>Mouse anti-SARS-CoV-2 S1 monoclonal antibody (MAB105403, R&D systems, 1:2,000),<br>These products were validated by putting negative controls of the antigens in western blotting. We verified that the bands detected at appropriated sizes were not observed in the samples from the antigen-negative cells.<br><br>Rabbit anti-GAPDH polyclonal antibody (10494-1-AP, Proteintech, 1:5,000). This antibody has been tested as stated on the supplier's website in human placenta tissue, Raji cells, HepG2 cells, K-562 cells, mouse heart tissue, A549 cells, PC-13 cells, arabidopsis whole plant tissue, corn whole plant tissue, mouse brain tissue, HEK-293 cells, HeLa cells, rat brain tissue, mouse skin tissue, RAW 264.7 cells, C6 cells, Jurkat cells, NIH/3T3 cells<br><br>Horseradish peroxidase (HRP)-conjugated Goat anti-rabbit IgG antibody (Cell Signaling, 7074, 1:5,000),<br>HRP-conjugated anti-mouse IgG antibody (Cell Signaling, 7076, 1:5,000).<br>Alexa Fluor647 Goat Anti-Human IgG secondary Ab (1.5 mg/ml) (Jackson Immunoresearch)<br>anti-GM130[EP892Y]-cis-Golgi (ab52649, 1:250)<br>phalloidin 647 (Thermofisher, A22287, 1:1000)<br>Goat anti-Rabbit IgG (H+L) Cross-Adsorbed Secondary Antibody, Alexa Fluor 594 (Thermofisher, A-11012, 1:1,000)<br>Goat anti-Mouse IgG (H+L) Cross-Adsorbed Secondary Antibody, Alexa Fluor 488 (Thermofisher, A-11001, 1:1,000)<br>These products were thoroughly validated by the providers and work optimally with the protocol for Western blotting or FACS or IF, as described in the Method section. We ensured the results are accurate and reproducible.<br><br>Casirivimab and Imdevimab were generated as described in the Method section. The nucleotide sequences of their templates were verified by Sanger sequencing. |
| --- | --- |

# Eukaryotic cell lines

Policy information about cell lines

| Cell line source(s) | HEK293T cells (a human embryonic kidney cell line; ATCC CRL-3216)<br>HOS-ACE2/TMPRSS2 cells (HOS cells stably expressing human ACE2 and TMPRESS2) (Saito et al., Nature, 2021)<br>VeroE6/TMPRSS2 cells [an African green monkey (Chlorocebus sabaeus) kidney cell line; JCRB1819]<br>Calu-3 cells (a human lung epithelial cell line; ATCC HTB-55 or a gift from Paul Lehner)<br>H1299 a gift from Simon Cook<br>HeLa-ACE2 a gift from James Voss<br>CaCo2 a gift from Ian Goodfellow<br>A549-ACE2/TMPRSS2 from Massimo Palmarini<br>293T (CRL-3216)<br>293T-ACE2/TMPRSS2, 293T-ACE2ΔTMPRSS2, 293T-TMPRSS2, 293T-GFP11 and Vero -GFP1-10 were provided by Leo James<br>Vero-ACE2/TMPRSS2 from Emma Thomson<br>ExpiCHO cells from Davide Corti<br>A549-ACE2-TMPRSS2-based luminescent reporter cells from Nick Matheson |
| --- | --- |
| Authentication | HEK293T cells, H1299 and Calu-3 cells were authenticated by ATCC.<br>VeroE6/TMPRSS2 cells was authenticated by JCRB Cell Bank.<br>HOS-ACE2/TMPRSS2 cells Saito et al., Nature, 2021) was not authenticated. |
| Mycoplasma contamination | All cell lines were regularly tested for mycoplasma contamination by using PCR and were confirmed to be mycoplasma-free. |
| Commonly misidentified lines<br>(See ICLAC register) | No commonly misidentified cell lines were used. |

# Human research participants

Policy information about studies involving human research participants

| Population characteristics | Vaccine sera were collected from fourteen vaccinees four weeks after the second vaccination with BNT162b2 (Pfizer-BioNTech) (average age: 46, range: 38-55, 21% male),<br><br>ten vaccinees four weeks after the second vaccination with mRNA-1273 (Moderna) (average age: 26, range: 21-37, 40% male),<br><br>nine vaccinees 4-6 weeks after the second vaccination with CoronaVac (Sinovac) (44% male).<br><br>Delta convalescent Delta sera were collected from ten vaccine-naive individuals who had infected with Delta variant (B.1.617.2 or AY.29) (average age: 46, range: 22-63, 70% male). |
| --- | --- |
| Recruitment | The voluntary donors were recruited at Kyoto University (BNT162b2 and mRNA-1273 vaccinees),<br><br>Universidad San Francisco de Quito (CoronaVac vaccinees), |

Kuramochi Clinic Interpark (Delta convalescents)

regardless of age, sex, gender, race, ethnicity, or other characteristics. Written informed consent was obtained from the voluntary donor.

Community vaccinated participants were also recruited by the NIHR BioResource

| Ethics oversight | All protocols involving specimens from human subjects recruited at |
| --- | --- |
| | Kyoto University, |
| | Kuramochi Clinic Interpark |
| | Universidad San Francisco de Quito |
| | were reviewed and approved by the Institutional Review Boards of |
| | Kyoto University (approval ID: G0697), |
| | Kuramochi Clinic Interpark (approval ID: G2021-004) |
| | Universidad San Francisco de Quito (approval ID: CEISH P2020-022IN), and the Ecuadorian Ministry of Health (approval IDs: MSP-CGDES-2020-0121-O and MSP-CGDES-061-2020). |
| | Ethical approval for study of vaccine elicited antibodies in sera from vaccinees was obtained from the East of England – Cambridge Central Research Ethics Committee Cambridge (REC ref: 17/EE/0025). |
| | Ethical approval for use of human tissue for measurement of ACE2 and TMPRSS2 levels was obtained as follows: the samples of human upper and lower airways were obtained from the lungs of a multiorgan donor whose lungs were deemed unsuitable for clinical transplantation and after their next of kin consented to their use in research. The studies using human donor lungs tissue were approved by National Research Ethics Committee (NREC) 16/NE/0230. |

Note that full information on the approval of the study protocol must also be provided in the manuscript.

# Flow Cytometry

## Plots

Confirm that:

☒ The axis labels state the marker and fluorochrome used (e.g. CD4-FITC).

☒ The axis scales are clearly visible. Include numbers along axes only for bottom left plot of group (a 'group' is an analysis of identical markers).

☒ All plots are contour plots with outliers or pseudocolor plots.

☒ A numerical value for number of cells or percentage (with statistics) is provided.

## Methodology

| Sample preparation | HEK293 cells were cotransfected with 400 ng of D614G S or D614G/P681R expression plasmids and 400 ng pDSP1-7 using TransIT-LT1 (Takara, Cat# MIR2300). |
| --- | --- |
| Instrument | FACS Canto II instrument (BD Biosciences) |
| Software | FlowJo |
| Cell population abundance | n/a (cell sorting was not conducted) |
| Gating strategy | Extended Data Fig. 8.<br>The cells were gated in the FSC/SSC plot, then the mean fluorescence intensity of surface S protein (APC) was measured. The boundary between S-positive and S-negative was defined by using the cells that the S-expression plasmid was not transfected (i.e., S-negative cells). |

☒ Tick this box to confirm that a figure exemplifying the gating strategy is provided in the Supplementary Information.

