## [Peer Review File · Nature]

Manuscript Title: Altered TMPRSS2 usage by SARS-CoV-2 Omicron impacts tropism and fusogenicity

Reviewer Comments & Author Rebuttals

Reviewer Reports on the Initial Version:

Referee #1 (Remarks to the Author):

This a solid study with respect to the infection properties of Omicron, given the short time-frame involved.

the data are consistent with less cleavage relative to Delta, although the substantially lowered expression of Omicron are a concern. Can the authors explain this phenomenon better? Fig 2b needs better annotation (S1 vs S2). The relatively high infectivity of H1299 cells is interesting, and should be expanded on or explained better.

It is unclear how modeling is useful in light of the recently released cryoEM structure of Omicron S, but is well done and analysis of the S1/S2 region is informative; however there appears to be a typo in the Delta figure (P681H/R)

The relative differences re TMPRSS2 expression are interesting but are not clearly presented and are underdeveloped, the authors should discuss or address more the use of other TTSPs in addition to TMPRSS2. TMPRSS2 is not a “receptor”

Studies on antiviral drugs are a useful addition.

Referee #2 (Remarks to the Author):

This submission provides important information on the omicron VOC. First, the report presents structural models for the omicron spike and its RBD interactions with ACE2. Next, the paper demonstrates that omicron is resistant to vaccine-induced antisera but less so after booster vaccination. The report also presents very much expected yet valuable findings that omicron is

sensitive to molnupiravir and remdesivir. Finally, the paper provides data on omicron entry, using pseudo virus transduction assays and some live virus infections, to demonstrate that omicron is less dependent on TMPRSS2 than previous delta VOC, and that omicron spikes induce less cell-cell fusion than previous D614G spikes.

The data showing reduced omicron neutralization by vaccine sera and clinical mAbs are valuable, convincing, and are consistent with many similar reports addressing this same question. However, there are concerns with several other aspects of the report, including the MD simulations, the interpretations of the PV transductions and cell-cell fusions, and the discussion of the findings in broader context. Specific concerns are itemized.

1. It is understood that the field moves very fast; a preprint posted Dec 21 doi: <https://doi.org/10.1101/2021.12.19.473380> shows cryoEM structures of omicron spike and spike-ACE2 complexes. These cryoEM structures do not appear to cohere fully with the structures predicted by molecular dynamics simulations. Discrepancies should be addressed.
2. The predicted exposed furin cleavage site is not consistent with the biochemical data showing reduced S1-S2 cleavage. This raises some questions, possibly regarding the MD simulations. This conundrum should be addressed.
3. Fig. 2; how are PV transduction data interpreted, given that there are two variables, one being lower omicron spike densities on the PVs, and the other being less omicron spike cleavage relative to delta? Which of these two variables might account for differential omicron PV transduction levels?
4. Fig 2; should have S1-S2 vs N or M western blots for authentic live viruses (this appears as a gap in the data). This is important because the secreted virions may not have the same spike cleavage extents as found in producer cell lysates.
5. Fig. 3; the relevant spike expression in the analysis of cell-cell fusion is plasma-membrane spike. Fig 2b western blot data cannot be used to infer similar cell surface levels for D614G and omicron spikes. Abundant uncleaved spikes in omicron cell lysates can come from failure of spikes to exit the ER.
6. In the Discussion section it is stated that “Recent findings from Hong Kong suggest higher replication in ex vivo lung tissue as compared to upper airway tissue for Omicron, but not for Delta (HKU website).” However the HKU site, dated 15 Dec 2021, states the opposite, here from the HKU site “The researchers found that Omicron SARS-CoV-2 infects and multiplies 70 times faster than the Delta variant and original SARS-CoV-2 in human bronchus, which may explain why Omicron may transmit faster between humans than previous variants. Their study also showed that the Omicron infection in the lung is significantly lower than the original SARS-CoV-2, which may be an indicator of lower disease severity.” This concern should be sorted out; at present it is not clear how the authors can claim that “a mechanism for omicron's poorer replication in the ex vivo tissue can be derived

from our findings regarding reduced Omicron spike cleavage and poor utilisation of TMPRSS2 dependent plasma membrane fusion by Omicron spike.”

Referee #3 (Remarks to the Author):

Meng et al. study the Omicron S protein and make several useful observations.

1. Consistent with other reports, they show that Omicron dramatically evades sera from vaccinated individuals and, as well as Regeneron’s antibody cocktail.
2. They further show a clear difference between initial vaccination with the Ad-vectored ChAdOx-1 vaccine and with the mRNA-LNP-delivered BNT162b2, and that a boost with the latter can greatly enhance Delta and Omicron
3. They confirm that two nucleoside analogues remain effective against Omicron
4. They observed – unexpectedly – that the Omicron furin site is less efficient – more similar to Wuhan-D614G than to Delta for example, in both cells and on pseudovirus
5. They observed much less incorporation of Omicron S into pseudoviruses
6. They observe a lower pseudovirus infectivity on Calu3 and lung organoids, but comparable levels of infection on H1299
7. They observe that H1299 have markedly lower TMPRSS2 and ACE2 than Calu3
8. They describe the distribution of ACE2 and TMPRSS2 upper and lower respiratory tract in specialized cells.
9. They show that syncytia formation is lower with Omicron than with Delta in the presence or absence of TMPRSS2

This paper is a bit of a grab bag of facts and factoids about Omicron. The strongest data is Fig. 1 – vaccine and antibody efficacy – are reported by several groups. Also interesting is the observation of relatively inefficient cleavage of Omicron S (Fig. 2). Unfortunately the most novel conclusions (Figs. 3 and 4) are also the weakest. Specifically, assertions pertaining to the role of TMPRSS2 here are unsubstantiated and speculative.

To draw this out, the authors show quite vividly that while the Omicron S protein expresses comparable to Delta and Wuhan-D614G (Fig 2E), it is not efficiently cleaved at the S1/S2 boundary (Fig 2B and 2E, like Wuhan but unlike Delta), and it poorly incorporates into the pseudovirion (Fig. 2B; As noted below, these observations render Fig. 4 useless). They authors observe that

pseudoviruses with the Omicron S protein infect lung organoids and Calu-3 less efficiently than those with Delta S, but Delta and Omicron pseudoviruses enter comparably on H1299 cells.

At this point the authors focus on the presence of TMPRSS2 as the relevant variable distinguishing H1299 cells from Calu-3 cells, but unfortunately without much solid evidence. Fig. 3A shows lower TMPRSS and lower ACE2 in H1299 compared with Calu 3 cells. Of course many genes (attachment factors, other proteases including other TMPRSS-family proteins and other endosomal proteases) could distinguish the entry processes of these cells. Fig. 3B then shows Wuhan-D614G and Delta infect ACE2-expressing 293T cells more efficiently in the presence of TMPRSS2. In contrast, Omicron is unaffected, suggesting that Omicron depends less on TMPRSS2. Many obvious controls (cathepsin inhibitors, camostat, Omicron with a Delta furin-cleavage region) are missing. An analysis of cell type distribution does not really add here, in large part because ACE2 and TMPRSS2 levels are coordinated together, with more of everything in the lower respiratory tract.

As mentioned, the data of Fig. 2 make Fig. 4 uninterpretable and, well, meaningless. Much lower incorporation into pseudovirions probably means the surface expression of the Omicron S protein is lower, and, even if not, its furin cleavage efficiency in cells is much lower. So the simplest inference one can make from Fig. 4 is that if you have less functional S protein, you observe less syncytia. The absence of any controls (cathepsin L inhibitor? TMPRSS2 inhibitor?, revert furin site? Lower pH as described for SARS-CoV-1 in Li et al, 2003 Nature?) makes this figure even weaker.

In short, what the authors are trying to imply is that Omicron is less dependent on Tmprss2 (possibly but not demonstrated. Note a similar point has been made for SARS-CoV-1, and the absence of a furin site has been decidedly implicated, see Ou et al. Plos Path 2021). They are then trying to say that this difference in dependence makes Omicron replicate more efficiently in the upper respiratory tract (unshown here) because there is a difference in TMPRSS2 expression in some cells (no persuasive correlation).

Referee #4 (Remarks to the Author):

Meng and colleagues report on Omicron sensitivity to monoclonals and serum as well as some of the properties of the virus in in vitro assays. They start by modeling the Omicron spike and comparing it to delta. They then use pseudoviruses to test the activity of a couple of monoclonal antibodies that are in clinical use and plasma and live virus to test the sensitivity against redemsevir and molnupiravir. The results are very much as expected and consistent with online reports by other groups. Although they fail to emphasize it in the text, it is clear from the data that boosting makes an important difference and this should be emphasized for public health reasons.

They then examine spike in pseudoviruses produced by transfection in vitro and compare to controls. They report less spike and less cleaved spike. Its hard to know what to make of this and whether it is relevant to real virus with real virus data. Similarly, the experiments on entry show heterogeneous results and are difficult to interpret in the absence of a measure of real virus production as opposed viral RNA in supernatants as reported.

Cells transfected with ACE2 and TMPRSS2 were used to probe entry requirements and the data indicate that Omicron is less sensitive to the absence of TMPRSS2 than delta or Wuhan. But the difference for those 2 strains is less than 2 fold. Its very hard to know what to make of this data with such small differences. Nevertheless, based on expression analysis they want to make a case that Omicron is less able to infect the lower airway.

Finally in a transfected cell fusion system they see less fusion with Omicron. But the differences are small.

Overall the manuscript reads as though it was very rushed. It is difficult to read in part because they frequently fail to state how they are doing the experiments or emphasize the in vitro nature of many of the experiments and detail the many many caveats with the experiments.

Problems:

1. In the first 2 paragraphs there is what is largely a model-based discussion of Omicron structure referring to Supplementary figures S1 and S2. The conclusions are based on modeling and not structural data. However, the text makes is sound as if its actual data. They should alter the text to emphasize right up front that they are talking about models which may need to be corrected when structural data becomes available.
2. The nature of the clinical cohorts and when the serum samples were collected should be clearly stated in the text. This information is in the Fig legend but is difficult to decipher.
3. They only measure a very limited number of the monoclonals that are in clinical use or will be in clinical use. Moreover, these assays are pseudovirus assays and the caveat should be noted more clearly.
4. They need to emphasize the boosting results
5. The measures of pseudovirus spike expression are made in transfected pseudoviruses and the relevance to real virus is not clear.
6. As in #4 for measures of S protein cleavage

7. The experiments on entry again with pseudovirus show heterogeneous results that are difficult to interpret. These need to be done with real virus on real lung cells and they need to measure real virus production as opposed to supernatant RNA.

8. The ACE2 and TMPRSS2 experiments suffer from the same caveats as do the cell fusion experiments.

Referee #5 (Remarks to the Author):

In this paper, the authors present an important and timely analysis of the omicron variant's immune response, cell-cell fusion ability, and infectivity using pseudovirus assays. Main conclusions: (1) Omicron exhibits significant evasion of vaccine elicited antibodies (also supported by preprint of Veesler & Corti at least); (2) Omicron demonstrates significantly lower replication in lung cells compared to Delta (also supported by Hong Kong preprint), (3) Omicron's fusogenicity is reduced despite TMPRSS2 expression, indicating that something inherent to the mutations that is reducing protease priming. These findings together indicate that Omicron has significant immune evasion properties and different tissue tropism.

I was asked to focus comments mainly on the simulation components of the work. The authors present models of the Omicron spike modeled off of 7A94, one of the earlier spike structures (I believe representing the Wuhan strain) from the group of Gamblin. They state that it contains 1146 residues but should clarify in the main text that this is for one chain of the trimer. The full spike model obviously contains 3X that value. Choosing the model the loops from Modeler is fine although some of the regions not resolved in 7A94 now have been resolved by several other groups, so I would have chosen to graft those regions or would have started from a different baseline structure.

The authors must specify which disulfides were present in the simulated constructs, as there are several missing disulfide bonds from the un-resolved regions that will make a difference in the structural dynamics.

However, another significant shortcoming of the work is that it seems the authors did not include glycosylation. The choice to omit such is not justified at this point. All data that has been published indicates that the glycans are essential to include. The reason for this is that: (1) we know the molecular composition of the N- and O-linked glycans from mass spec / glycomics (from the work of Crispin, eg., and Azadi, and others), (2) we know these sites are nearly fully occupied (many

published studies), we know that the glycans impact the degree of shielding as well as contacts with ACE2 (and similarly, accurate pictures of the ACE2-spike interactions should also include glycans on ACE2, which have been shown to make contact to the spike in other published studies) (see work of Amaro, Hummer, Fadda, at least). Moreover, there are many fully glycosylated models of the spike that have already been vetted by the community and shared widely, so it makes very little sense to not use those as a basis into which mutations from Omicron are added. As it stands, the current models of the spike protein included here are in my opinion unfit for publication.

Another significant issue is that the authors run single copy MD simulations of 100 ns in length. The length is quite short – and I can appreciate this because of the need to push out this data quickly – but including only one copy (no replicas) is widely considered to not be ‘best practice’ in terms of MD methods. Replicas must be considered, and in fact most disciplinary journals in computational chemistry / biophysics now require that even before such papers are sent for review. In this case, the authors run one copy of MD and then attempt to draw conclusions about contacts and so forth. With only one copy the authors simply cannot state that the data supports the conclusion because the data is too preliminary (replicas must be run in order to provide error bars on values provided).

Furthermore, SI Figure 1 attempts to draw conclusions about contacts with > 50% persistence. This analysis simply must contain glycans. Without glycans many of these computed values will be erroneously / not reflecting experimental reality. On top of that, the dynamics are probably too short (100ns) to draw reasonable conclusions on this.

I wish I could be more positive about the computational efforts attached herein, but as described there are significant shortcomings that in my expert opinion render the conclusions drawn as invalid, and/or too premature for publication.

Otherwise this seems a very timely and important contribution to the literature. I appreciate that all this work is an enormous effort made under intense pressure and around the holidays. I thank all the authors for their efforts in this regard.

Author Rebuttals to Initial Comments:

Referee #1 (Remarks to the Author):

This a solid study with respect to the infection properties of Omicron, given the short time-frame involved.

1. the data are consistent with less cleavage relative to Delta, although the substantially lowered expression of Omicron are a concern. Can the authors explain this phenomenon better?

Response: we thank the reviewer for these two related comments. Our updated work with western blots using live virus shows that indeed cellular spike is inefficiently cleaved, as observed with PV. In live virions we see that virion associated spike appears lower for Omicron, though also slower to express, likely due to less efficient omicron entry in the T2 high producer VeroE6 A2T2 cells. However, replication in the human nasal 3D system is similar for live Omicron versus Delta. This is likely because Omicron spike has higher affinity for ACE2 – we now show this in two ways experimentally (BLI with immobilised Spike RBD, and a second full spike expression system). Lower expression of higher affinity spike might be an antibody evasion mechanism and we know this from HIV-1, which has only 7-13 trimers on its surface – enough for efficient infection.

Lower virion expression in PV has been observed by other groups, and as the reviewer points out, the lower virion spike expression in the PV still enables Omicron spike to achieve similar infectivity to Delta in H1299 (low in TMPRSS2), and indeed other cells where TMPRSS2 is not overexpressed or highly expressed (eg HeLa-ACE2).

The relatively high infectivity of H1299 cells is interesting, and should be expanded on or explained better.

Response: we extend the data for efficient infection in H1299 cells by showing efficient infection in TMPRSS2 low cells as follows:

(i) we show a similar PV infectivity between Omicron and Delta in cells where TMPRSS2 is not overexpressed. Upon overexpression / high endogenous expression of T2 we see a difference in entry efficiency between Omicron and Delta.

(ii) we show live virus infection of human nasal epithelial cultures (hNEC) with an air liquid interface and demonstrate similar infection kinetics for Omicron and Delta.

(iii) We also demonstrate the mechanism behind the data by exploring sensitivity to drugs that block the endosomal (non TMPRSS2 dependent) pathway. We show that Omicron is more dependent than Delta on the endosomal route as expected. These data are consistent with the fact that Omicron does not use the TMPRSS2 mediated route efficiently and has increased its reliance on cathepsin dependent endosomal entry.

Fig 2b needs better annotation (S1 vs S2).

Response: we have amended the annotation

2. It is unclear how modeling is useful in light of the recently released cryoEM structure of Omicron S, but is well done and analysis of the S1/S2 region is informative; however there appears to be a typo in the Delta figure (P681H/R).

Response: we thank the reviewer for this. Indeed there are CryoEM structures now (Subramaniam and Trono labs). However, as pointed out, the cleavage regions (S1/S2 and S2') are not resolved in CryoEM and therefore MD would be a useful tool to explore this. We have removed the MD part as we wanted to do longer, more complex simulations with glycosylated structures and this will take considerable additional time. We therefore decided after discussion with the editor to remove this section from a paper that is already extensive and long.

3. The relative differences re TMPRSS2 expression are interesting but are not clearly presented and are underdeveloped, the authors should discuss or address more the use of other TTSPs in addition to TMPRSS2.

Response: we have now more thoroughly explored TMPRSS2 expression in cells and tissues and added considerably more data from live virus replication experiments on organoids and cell lines as outlined in response to point 1 above. We have also related the level of TMPRSS2 RNA transcripts with replication/entry by live virus in cell lines. Additionally we have done experiments with TMPRSS2 KO and overexpression systems in both ACE2 high and low contexts to show Omicron does not efficiently use TMPRSS2 for entry. This finding is confirmed further by showing lower sensitivity of Omicron to camostat and higher sensitivity of Omicron to cathepsin inhibitors. Finally as suggested, we have also discussed TTSPs more broadly in the discussion.

Specific data on TMPRSS2 demonstrating impact on Omicron

Importantly we show with the TMPRSS2 KO versus overexpressing cells that this increase in TMPRSS2 leads to increased entry by Delta but not omicron PV. We show that the level of ACE2 expression can influence the effect size of TMPRSS2 expression on enhancement of entry for Delta and Omicron. The impact of T2 expression is far greater when ACE2 is limiting/not overexpressed.

We also use drugs that block TMPRSS2 dependent and independent pathways to show that Omicron is more sensitive to blockade of the endocytic pathway and Delta more sensitive to blockade of plasma membrane fusion. These drug data done in both PV and live virus provide orthogonal support for the shift in tropism away from TMPRSS2 expressing cells.

In addition we have new data showing reduced replication in CaCo-2 gut cells and reduced entry in gall bladder organoids (a known target cell type for SARS-CoV-2) that express high levels of TMPRSS2. These are important data as persistence of SARS-CoV-2 in GI tissues over months has been reported (Nussensweig lab, Nature 2021) and a tropism switch away from such cells in the GI tract could reduce persistence.

4. TMPRSS2 is not a “receptor”

Response: yes we made a mistake here using the word ‘receptor’, clearly it is a protease and we have amended the text. We now refer to TMPRSS2 as a co-factor

5. Studies on antiviral drugs are a useful addition.

Response: we thank the reviewer for appreciating the importance of antiviral drugs targeting non spike proteins where mAb efficacy has been compromised.

Referee #2 (Remarks to the Author):

This submission provides important information on the omicron VOC. First, the report presents structural models for the omicron spike and its RBD interactions with ACE2. Next, the paper demonstrates that omicron is resistant to vaccine-induced antisera but less so after booster vaccination. The report also presents very much expected yet valuable findings that omicron is sensitive to molnupiravir and remdesivir. Finally, the paper provides data on omicron entry, using pseudo virus transduction assays and some live virus infections, to demonstrate that omicron is less dependent on TMPRSS2 than previous delta VOC, and that omicron spikes induce less cell-cell fusion than previous D614G spikes.

The data showing reduced omicron neutralization by vaccine sera and clinical mAbs are valuable, convincing, and are consistent with many similar reports addressing this same question. However, there are concerns with several other aspects of the report, including the MD simulations, the interpretations of the PV transductions and cell-cell fusions, and the discussion of the findings in broader context. Specific concerns are itemized.

1. It is understood that the field moves very fast; a preprint posted Dec 21 [doi:https://doi.org/10.1101/2021.12.19.473380](https://doi.org/10.1101/2021.12.19.473380) shows cryoEM structures of omicron spike and spike-ACE2 complexes. These cryoEM structures do not appear to cohere fully with the structures predicted by molecular dynamics simulations. Discrepancies should be addressed.

2. The predicted exposed furin cleavage site is not consistent with the biochemical data showing reduced S1-S2 cleavage. This raises some questions, possibly regarding the MD simulations. This conundrum should be addressed.

Response: we thank the reviewer for these two related points. We have removed the MD part as we wanted to do longer, more complex simulations with glycosylated structures following peer review comments and this will take considerable additional time.

3. Fig. 2; how are PV transduction data interpreted, given that there are two variables, one being lower omicron spike densities on the PVs, and the other being less omicron spike cleavage relative to delta? Which of these two variables might account for differential omicron PV transduction levels?

Response: We show that despite lower spike levels and lower cleavage, the spike has higher affinity for ACE2. This property allows Omicron to compensate for lower spike levels in cells such as human nasal epithelia, H1299, Hela-ACE2 and 293TdeltaTMPRSS2 cells where the endosomal route is dominant (low TMPRSS2 and S1/S2 cleavage not critical as cathepsins can cleave S1/S2 in endosomes). The cleavage impairment manifests as lower replication in cells where TMPRSS2 is expressed (CaCo-2, Calu-3, Hela-A2T2 overexpressing cells). We show that the differential PV transduction levels are also due to presence of TMPRSS2 (as found in lung organoids, gallbladder organoids, CaCo-2 gut cells and Calu-3 lung cells as shown in the revised paper) which gives Delta an entry advantage over Omicron.

4. Fig 2; should have S1-S2 vs N or M western blots for authentic live viruses (this appears as a gap in the data). This is important because the secreted virions may not have the same spike cleavage extents as found in producer cell lysates.

Response: we now provided western blots of live virions and the cell lysates from live virus infected cells.

We have also provided IF images of cells infected with live virus (stained for spike) showing that the distribution of virus does not appear to differ between Omicron and Delta.

5. Fig. 3; the relevant spike expression in the analysis of cell-cell fusion is plasma-membrane spike. Fig 2b western blot data cannot be used to infer similar cell surface levels for D614G and omicron

spikes. Abundant uncleaved spikes in omicron cell lysates can come from failure of spikes to exit the ER.

Response: we agree with this point and have provided flow cytometry data to show surface levels of spike. Indeed Omicron spike does reach the plasma membrane to a similar extent as delta spike and therefore fusion differences are real. We now also provide data on infection focus size differences between Omicron and Delta in semi solid media where the dominant route of virus spread is cell-cell.

6. In the Discussion section it is stated that “Recent findings from Hong Kong suggest higher replication in ex vivo lung tissue as compared to upper airway tissue for Omicron, but not for Delta (HKU website).” However the HKU site, dated 15 Dec 2021, states the opposite, here from the HKU site “The researchers found that Omicron SARS-CoV-2 infects and multiplies 70 times faster than the Delta variant and original SARS-CoV-2 in human bronchus, which may explain why Omicron may transmit faster between humans than previous variants. Their study also showed that the Omicron infection in the lung is significantly lower than the original SARS-CoV-2, which may be an indicator of lower disease severity.” This concern should be sorted out; at present it is not clear how the authors can claim that “a mechanism for omicron's poorer replication in the ex vivo tissue can be derived from our findings regarding reduced Omicron spike cleavage and poor utilisation of TMPRSS2

dependent plasma membrane fusion by Omicron spike.”

Response: we apologise for the misunderstanding. Our data are entirely in keeping with the HKU data. We misworded the text and stated the findings the wrong way around. The text now reads: “Recent findings from Hong Kong suggest lower replication in ex vivo lung tissue as compared to upper airway tissue for Omicron, but not for Delta (HKU website).”

Referee #3 (Remarks to the Author):

Meng et al. study the Omicron S protein and make several useful observations.

1. Consistent with other reports, they show that Omicron dramatically evades sera from vaccinated individuals and, as well as Regeneron’s antibody cocktail.

2. They further show a clear difference between initial vaccination with the Ad-vectored ChAdOx-1 vaccine and with the mRNA-LNP-delivered BNT162b2, and that a boost with the latter can greatly enhance Delta and Omicron

3. They confirm that two nucleoside analogues remain effective against Omicron

4. They observed – unexpectedly – that the Omicron furin site is less efficient – more similar to Wuhan-D614G than to Delta for example, in both cells and on pseudovirus
5. They observed much less incorporation of Omicron S into pseudoviruses
6. They observe a lower pseudovirus infectivity on Calu3 and lung organoids, but comparable levels of infection on H1299
7. They observe that H1299 have markedly lower TMPRSS2 and ACE2 than Calu3
8. They describe the distribution of ACE2 and TMPRSS2 upper and lower respiratory tract in specialized cells.
9. They show that syncytia formation is lower with Omicron than with Delta in the presence or absence of TMPRSS2

This paper is a bit of a grab bag of facts and factoids about Omicron. The strongest data is Fig. 1 – vaccine and antibody efficacy – are reported by several groups. Also interesting is the observation of relatively inefficient cleavage of Omicron S (Fig. 2). Unfortunately the most novel conclusions (Figs. 3 and 4) are also the weakest. Specifically, assertions pertaining to the role of TMPRSS2 here are unsubstantiated and speculative.

Response: we thank the reviewer for the points regarding neutralisation resistance.

Regarding figures 3 and 4 we have now added a considerable body of new data in support of the novel finding that Omicron has altered tropism moving away from TMPRSS2 mediated PM fusion towards endosomal entry. Indeed we now provide data in primary cells/tissue using live virus, as well as PV systems.

To draw this out, the authors show quite vividly that while the Omicron S protein expresses comparable to Delta and Wuhan-D614G (Fig 2E), it is not efficiently cleaved at the S1/S2 boundary (Fig 2B and 2E, like Wuhan but unlike Delta), and it poorly incorporates into the pseudovirion (Fig. 2B; As noted below, these observations render Fig. 4 useless).

Response: we thank the reviewer for the positive comment and in response to the incorporation comment now provide flow cytometry data to show that in fact Omicron spike is well expressed at the cell surface – therefore fig 4 is still valid. We now also provide western blots of live virus and live virus infected cell lysates. These show similar results as the PV blots, namely that cleavage is reduced in Omicron spike. However, there is reasonable incorporation of spike into virions to an extent that replication in human nasal epithelial cultures with air liquid interface is similar for Omicron v Delta.

In addition the spike PV for both variants have similar infectivity in cells where TMPRSS2 is not expressed. The higher ACE2 affinity likely compensates for lower spike levels in virions, or alternatively there may already be enough spike on Omicron virions to achieve efficient infection.

They authors observe that pseudoviruses with the Omicron S protein infect lung organoids and Calu-3 less efficiently than those with Delta S, but Delta and Omicron pseudoviruses enter comparably on H1299 cells.

At this point the authors focus on the presence of TMPRSS2 as the relevant variable distinguishing H1299 cells from Calu-3 cells, but unfortunately without much solid evidence.

Response: we now make a much stronger case supported by solid evidence that Omicron has shifted away from TMPRSS2 expressing cells, and extended further using TMPRSS2 overexpression and KO cells, as well as inhibitors of cathepsins and TMPRSS2 to show Omicron's route of entry has shifted towards endocytosis.

Fig. 3A shows lower TMPRSS2 and lower ACE2 in H1299 compared with Calu 3 cells. Of course many genes (attachment factors, other proteases including other TMPRSS-family proteins and other endosomal proteases) could distinguish the entry processes of these cells.

Response: Yes this is true, but the series of experiments presented in the revised manuscript offer firm evidence for a switch in entry preference for omicron to the endosomal route.

Fig. 3B then shows Wuhan-D614G and Delta infect ACE2-expressing 293T cells more efficiently in the presence of TMPRSS2. In contrast, Omicron is unaffected, suggesting that Omicron depends less on TMPRSS2. Many obvious controls (cathepsin inhibitors, camostat, Omicron with a Delta furin-cleavage region) are missing.

Response: We have now provided data in A2T2 overexpressing A549 lung cells treated with cathepsin inhibitors and camostat, showing that Omicron is more dependent on cathepsin mediated endosomal entry (green omicron PV and blue Delta PV), and that Delta is more dependent on TMPRSS2 mediated entry. We have performed both PV and live virus experiments with these drugs, with similar findings in both systems.

An analysis of cell type distribution does not really add here, in large part because ACE2 and TMPRSS2 levels are coordinated together, with more of everything in the lower respiratory tract.

Response: whilst indeed ACE2 and TMPRSS2 are part of an immune program, their levels do not necessarily co-ordinate given complexities of cellular gene expression. To that end we would like to clarify that the RNAseq data submitted do NOT show that both T2 and A2 are elevated in the lower tract, only that T2 is clearly higher. One cannot conclude that ACE2 is higher in lower airway as AT1/2 cells are an outlier with v high ACE2 and TMPRSS2 expression (of note club cells rather than AT1/2 cells are thought to be the primary cell type infected in the alveoli and club cells have relatively low ACE2 with high TMPRSS2). We have moved this figure to supplementary data. We provide qPCR data from cell lines and organoids in the revised paper, showing expression of ACE2 and TMPRSS2.

Additionally, we now provide our own tissue data showing ACE2 is expressed at similar and possibly higher levels in upper airway bronchial tissue by qPCR, and that TMPRSS2 is elevated in the lower parenchymal tissue.

As mentioned, the data of Fig. 2 make Fig. 4 uninterpretable and, well, meaningless. Much lower incorporation into pseudovirions probably means the surface expression of the Omicron S protein is lower, and, even if not, its furin cleavage efficiency in cells is much lower. So the simplest inference one can make from Fig. 4 is that if you have less functional S protein, you observe less syncytia.

Response: we now have data showing robust levels of surface spike expression of Omicron spike. Our data show that the cellular spike is less well cleaved whether we consider live virus or spike transfection, and therefore we agree that the inference from our data is that Omicron spike is suboptimally cleaved despite adequate surface expression, and therefore is associated with poorer fusion, as cleaved SARS-CoV-2 spike is the ideal substrate for TMPRSS2 priming of S2' and fusion peptide activation. This is nonetheless a very important point as it could impact pathogenesis. Interestingly, SARS-CoV-1 forms syncytia as shown by the Farzan lab (Nature 2003), and it can be cleaved at the cell surface by TMPRSS2, unlike SARS-CoV-2 which requires furin cleavage in a producer cell before it can be primed by TMPRSS2.

The absence of any controls (cathepsin L inhibitor? TMPRSS2 inhibitor?, revert furin site? Lower pH as described for SARS-CoV-1 in Li et al, 2003 Nature?) makes this figure even weaker.

Response: It is well accepted that TMPRSS2 facilitates fusion for SARS-CoV2– shown by us and other groups (Papa et al, PloS Pathogens 2020; Buchrieser et al, EMBO J 2020; Ou etl, PloS

Pathogens 2021). Our assay has already been validated in previous papers (Papa et al, PloS Pathogens 2020, Meng et al, Cell Reports 2021;Cattin-Ortola, Nature Comms 2021). These papers show that fusion is highly inefficient in the absence of TMPRSS2, possibly due to other proteases such as MMPs. We have now presented controls showing that if furin mediated cleavage of S1/2 is inhibited by CMK, cell-cell fusion is prevented, see supplementary figures. Further controls for the fusion assay include use of convalescent sera to prevent fusion in a dose dependent manner, indicating fusion is ACE2 dependent (supplementary figures). We do not think cathepsin inhibitors are a reasonable control for fusion experiments as Omicron does not induce fusion and cathepsin cannot substitute for TMPRSS2; therefore an inhibitor of cathepsin would not be useful in this experiment. Finally we would like to emphasise that the point of the experiment is that under conditions where delta forms syncytia, omicron does not. The low cleavage of Omicron spike is likely at least partly responsible, along with possible inefficient interaction of Omicron spike with TMPRSS2.

We also thank the reviewer for pointing out the paper: SARS-CoV-1 in Li et al, 2003 Nature, which identified ACE2 as the receptor for SARS-CoV-1. We could not find data on low pH in that paper though we have cited it in the discussion in relation to fusion and TMPRSS2 processing at the plasma membrane surface.

We have extended the data to show smaller infection focus size for Omicron versus Delta that we posit to be related to impaired fusogenicity.

In short, what the authors are trying to imply is that Omicron is less dependent on Tmprss2 (possibly but not demonstrated. Note a similar point has been made for SARS-CoV-1, and the absence of a furin site has been decidedly implicated, see Ou et al. Plos Path 2021).

Response: We feel that we have now addressed this comment with the additional experiments as outlined above on how TMPRSS2 manipulation in cell lines directly impacts entry of Delta but not Omicron, as well as drug inhibitor experiments for endocytosis v plasma membrane fusion. We thank the reviewer for pointing out the paper on importance of the furin cleavage site for entry pathway by Ou et al, which is now cited. An interesting parallel with SARS-CoV-1 is indeed becoming apparent in terms of viral entry pathway preference for endocytosis, partly associated with the degree of S1/2 cleavage that can be achieved. We thank the reviewer for highlighting this parallel and have now included it in the discussion.

They are then trying to say that this difference in dependence makes Omicron replicate more efficiently in the upper respiratory tract (unshown here) because there is a difference in TMPRSS2 expression in some cells (no persuasive correlation).

Response: we thank the reviewer for this very valid point. In response, we now show in nasal primary cultures (hNEC) that Omicron has similar replication to Delta. Our RNA seq and qPCR data show lower TMPRSS2 expression in upper airway compared to lower airway. This is in stark contrast to lower replication in lower airway cells, gut cells and gallbladder cells where TMPRSS2 is high. In our TMPRSS2+ versus TMPRSS2- cells we show enhancement for Delta but not Omicron in the presence of TMPRSS2. Inhibitor data using ED64 and camostat also support these conclusions regarding cell tropism. We believe these additional data address the concern expressed regarding linkage of Omicron replication with TMPRSS2. We agree with the reviewer that linking the findings of in vitro cell tropism to tissue tropism is more associative and therefore have altered the wording to reflect this, and we state that animal studies may help further. We have moved the RNAseq data to the supplementary material.

Referee #4 (Remarks to the Author):

Meng and colleagues report on Omicron sensitivity to monoclonals and serum as well as some of the properties of the virus in in vitro assays. They start by modeling the Omicron spike and comparing it to delta. They then use pseudoviruses to test the activity of a couple of monoclonal antibodies that are in clinical use and plasma and live virus to test the sensitivity against redemsevir and molnupiravir. The results are very much as expected and consistent with online reports by other groups. Although they fail to emphasize it in the text, it is clear from the data that boosting makes an important difference and this should be emphasized for public health reasons.

Response: we agree and thank the reviewer for evaluating this paper. We have now emphasised the importance of a third dose and the added breadth of neutralisation that it brings.

They then examine spike in pseudoviruses produced by transfection in vitro and compare to controls. They report less spike and less cleaved spike. Its hard to know what to make of this and whether it is relevant to real virus with real virus data.

Response: we have repeated cleavage analysis with live virions and cells infected with live virus. The results show a similar reduced cleavage for Omicron versus Delta that we see in the PV system. The amount of spike in live virus appears reduced for Omicron v Delta in both live and PV. The PV

difference does not translate to differences in infectivity in cell lines low in TMPRSS2, and this is likely due to the increased affinity of Omicron spike for ACE2 that we show (also confirmed by multiple other lab pre prints). Similarly, the live Omicron replicates as well as Delta in human nasal epithelial cell cultures. Greater spike abundance does not necessarily equate to higher infectivity. In vivo more glycoprotein makes the virus more visible to antibodies, and we make this point.

Similarly, the experiments on entry show heterogeneous results and are difficult to interpret in the absence of a measure of real virus production as opposed viral RNA in supernatants as reported.

The entry experiments, now including other cell lines, show that Omicron can achieve similar entry in some cells but not others, and that Omicron is disadvantaged in cells endogenously expressing or overexpressing TMPRSS2. The heterogeneity in entry across cell lines and organoids is explained by TMPRSS2 expression and that is a key message of the paper – tropism switch.

Cells transfected with ACE2 and TMPRSS2 were used to probe entry requirements and the data indicate that Omicron is less sensitive to the absence of TMPRSS2 than delta or Wuhan. But the difference for those 2 strains is less than 2 fold. Its very hard to know what to make of this data with such small differences. Nevertheless, based on expression analysis they want to make a case that Omicron is less able to infect the lower airway.

Response: we used transduced cells to overexpress TMPRSS2 and CRISPR for the TMPRSS2 knock out to enable a direct comparison of the impact of TMPRSS2 on entry in the same cell type. We have clarified this in the text. On checking the fold change we realise we made an error and the difference is 3.4x. This FC in a single round is significant and over multiple replication cycles could rapidly result in impact of greater than a log difference, as we show in live virus experiments in cells expressing TMPRSS2 (Caco2 cells, Calu-3 lung cells). Indeed when we do the same experiment in cells without overexpression of ACE2 the difference is much greater (Figure 3).

Finally in a transfected cell fusion system they see less fusion with Omicron. But the differences are small.

Response: with respect, we think the reviewer has misread the figure. The difference between Delta or WT and Omicron large (>10x).

Overall the manuscript reads as though it was very rushed. It is difficult to read in part because they frequently fail to state how they are doing the experiments or emphasize the in vitro nature of many of the experiments and detail the many many caveats with the experiments.

Response: we have now included more details of experimental methodology throughout the text and along with a section on caveats/limitations in the discussion.

Problems:

1. In the first 2 paragraphs there is what is largely a model-based discussion of Omicron structure referring to Supplementary figures S1 and S2. The conclusions are based on modeling and not structural data. However, the text makes it sound as if its actual data. They should alter the text to emphasize right up front that they are talking about models which may need to be corrected when structural data becomes available.

Response: we have amended the text and we no longer use MD in this manuscript.

2. The nature of the clinical cohorts and when the serum samples were collected should be clearly stated in the text. This information is in the Fig legend but is difficult to decipher.

Response: we apologise for the lack of clarity and now have a summary table for the longitudinal cohort, as well as more description in the text.

3. They only measure a very limited number of the monoclonals that are in clinical use or will be in clinical use. Moreover, these assays are pseudovirus assays and the caveat should be noted more clearly.

Response: The Regeneron cocktail is the most widely used mAb therapy in Europe and the US, hence the focus on that. Others have not been fully deployed as yet (AZ and GSK Sotrovimab), or have been withdrawn eg Lily. For these experiments live viruses were used. We will clarify this in text and legend.

4. They need to emphasize the boosting results

Response: indeed thank you yes we have now done this

5. The measures of pseudovirus spike expression are made in transfected pseudoviruses and the relevance to real virus is not clear.

Response: we have generated live virion western blot data and our new data for cell lysates infected with live virus reflects the PV system well. The reduced spike expression for omicron is somewhat apparent in live virus cell lysates as well, and could be due to a number of factors. We also show in live virions that there is also a reduction in total and cleaved spike for Omicron, reflecting the PV findings. To explore this we do IF of live virus in cells and it shows virus distribution is similar for Omicron and Delta. The mechanism for this warrants exploration, though we show that in nasal cultures and cell lines such as H1299, that the replication/entry of Omicron is similar to Delta. This may be because there is enough spike on the virion for infectivity and / or because of increased affinity for ACE2 that we show in the revised manuscript. Of note there is more cleaved spike in virions than in lysates, and this has been noted previously (Meng et al, Cell Reports 2021; Mlcochova et al, Nature 2021). This could be due to preferential incorporation of cleaved spike into virion associated membranes.

6. As in #5 for measures of S protein cleavage. Please see above response

7. The experiments on entry again with pseudovirus show heterogeneous results that are difficult to interpret. These need to be done with real virus on real lung cells and they need to measure real virus production as opposed to supernatant RNA.

Response: As mentioned above the heterogeneity is explained by TMPRSS2 expression, and we provide further evidence for this using primary cells and live virus. Figure 2 already showed live virus infection in Calu-3 lung cells. We have now done live virus in human nasal epithelial cultures and Caco-2 cells with measurement of infectious virus in supernatant.

8. The ACE2 and TMPRSS2 experiments suffer from the same caveats as do the cell fusion experiments.

Response: we have done ACE2 and TMPRSS2 experiments, as well as drug assays, in live virus as well as the PV system but will nonetheless make relevant caveats to the analyses presented, including fusion analysis.

Referee #5 (Remarks to the Author):

We have removed the MD section and therefore not responded to this review

Reviewer Reports on the First Revision:

Referee #2 (Remarks to the Author):

The resubmission is significantly improved. Amongst the improvements: (1) superior organization of text (2) removal of incomplete work on MD simulations (3) inclusion of several results documenting differential utilization of TMPRSS2 protease by VOCs.

Given that this is a second round of review, and given that the investigators have responded thoroughly to prior reviews, there are only a few remaining minor comments.

1. Effects of protease inhibitors on live VOC virus infections (Fig 3h): Relative lack of differences between delta and omicron in E64d suppression could arise because E64d blocks the common viral 3CL proteases that operate in viral replication. Suggest indicating limitations to using E64d in that the inhibitor has targets beyond the cathepsins that operate in entry.
2. TMPRSS2 levels and ACE2 levels: A common theme in the manuscript is with omicron failing to use TMPRSS2 for activation (at least not as much as delta), and with ACE levels relating somehow to this failure. For example, see line 399, :TMPRSS2 cofactor usage is impacted by ACE2 levels. One issue is that increasing TMPRSS2 suppresses ACE2 surface levels. The TMPRSS2 cleaves the ACE2. This appears to be a limitation in the study that might affect data interpretation and might warrant mention in the final paragraph.
3. Patterns of spike protein cleavage: Lines 385-387; proportion of cleaved spike was lower within cells relative to virions. The explanation here is that authentic virion spikes, incorporated at the ERGIC budding site, are cleaved as the virions secrete through exocytic pathways. This is different than the explanation provided.

Referee #3 (Remarks to the Author):

Meng et al. have responded to the reviewers' comments and in the process greatly improved the manuscript. The immune escape data were already robust. Now the impact of Tmprss2-dependence on cellular tropism is also persuasive. My confidence in the data is largely due to the inclusion of critical controls in Figure 3, and especially new Figures 3g and 3h, which are smoking guns for differential Tmprss2 dependence of the two S protein. They are also consistent with studies indicating that Tmprss2-dependence correlates with efficient furin-site cleavage. If me, I would make this link more forcefully since other groups have shown that transfer of a furin site to SARS1 increases tmprss2 dependence and removal of the furin site from SARS2 decreases tmprss2 dependence. Regardless, Figs 1-3 are well executed and very important.

Figure 4 remains less persuasive and disagree with its interpretation. To walk through it:

Fig 4a simply represents the assay used.

Fig. 4b is used to indicate that “[spike] expression achieve similar levels as Delta and Wuhan.” Not true. We are presented a bar chart representing the number of spike-expressing cells, indicating that transfection efficiencies were comparable. However this data cannot be used to state that expression is comparable. Show us the histograms! Observations from the previous version of the manuscript (now I think Supp Fig. 6A) indicate that the Omicron S alone expresses at much lower levels. I suspect we there will be fewer detectable S proteins per transfected cell than with delta or D614G. If so, the results of Fig. C and D do not say much except again that lower S protein expression leads to lower syncytia.

Fig 4e is less flawed but its interpretation is unpersuasive. Vero cells, I believe, are Tmprss2 low, which should favor Omicron over Delta (according to the persuasive Fig. 3). So again, Fig. 4e results may again be a simple consequence of lower surface expression of S. (Remember that the live virus buds from the ERGIC not the plasma membrane, so the amount of S protein at the plasma membrane, and incorporated in the retroviral PV, doesn't directly reflect actual virion production). Shouldn't the authors be arguing that syncytia formation limits the burst size of Delta? What are they actually arguing? To quote them, “We predicted that poor fusion would limit cell-cell spread of Omicron”. That is, the failure to form syncytia would be disadvantageous for Omicron spread. I might instead interpret these data to suggest that Omicron replicates more efficiently in TMPRSS2-low cells because it does not form syncytia that accelerate producer-cell death. Why is total virion production in the supernatant not measured?

Summarizing my concerns with Fig, 4: the data point to a model that the delta form syncytia because more functional spike is on the plasma membrane, even if Fig. 4b obfuscates that point. (I could be wrong – but we need the right flow data to know). Even so, we don't have any sense yet whether that is 'good' for Omicron (because syncytia accelerate cell death, limiting burst size) or, as the authors argue, because these syncytia model cell-to-cell transmission, 'good' for Delta. This latter interpretation would suggest that Delta would have higher viral loads in a low-Tmprss2 setting reflective of the upper respiratory track, contrary to the population transmission data.

Despite these concerns, the paper is important and generally well executed.

Caught one typo: E64D not ED64 line 283

Referee #4 (Remarks to the Author):

the manuscript is much improved by the additional experiments.

Referee #3

(Remarks to the Author)

Meng et al. have responded to the reviewers' comments and in the process greatly improved the manuscript. The immune escape data were already robust. Now the impact of Tmprss2-dependence on cellular tropism is also persuasive. My confidence in the data is largely due to the inclusion of critical controls in Figure 3, and especially new Figures 3g and 3h, which are smoking guns for differential Tmprss2 dependence of the two S protein. They are also consistent with studies indicating that Tmprss2-dependence correlates with efficient furin-site cleavage. If me, I would make this link more forcefully since other groups have shown that transfer of a furin site to SARS1 increases tmprss2 dependence and removal of the furin site from SARS2 decreases tmprss2 dependence. Regardless, Figs 1-3 are well executed and very important.

Response: we thank the reviewer for this invaluable comment. We indeed have now made the link between cleavage and TMPRSS2 usage and cited the furin cleavage transfer data to back this up.

Figure 4 remains less persuasive and disagree with its interpretation. To walk through it:

Fig 4a simply represents the assay used.

Response: no comment

Fig. 4b is used to indicate that “[spike] expression achieve similar levels as Delta and Wuhan.” Not true. We are presented a bar chart representing the number of spike-expressing cells, indicating that transfection efficiencies were comparable. However this data cannot be used to state that expression is comparable. Show us the histograms! Observations from the previous version of the manuscript (now I think Supp Fig. 6A) indicate that the Omicron S alone expresses at much lower levels. I suspect we there will be fewer detectable S proteins per transfected cell than with delta or D614G. If so, the results of Fig. C and D do not say much except again that lower S protein expression leads to lower syncytia.

Response: we agree with this criticism as % positive cells do not tell you much about expression per cell. We have now provided the histogram below. Interestingly the expression level is similar at the plasma membrane for the different spikes. We think the explanation is most likely that as TMPRSS2 is known to be vital for successful cell-cell fusion by SARS-CoV-2 S, and given the severe impairment of intracellular S1/2 cleavage efficiency (greater impairment than observed in virions), Omicron spike cannot be primed by TMPRSS2 to release the fusion peptide, despite being at the cell surface. This links with the previous comment from the reviewer on making clearer the relation between efficient cleavage and engagement of TMPRSS2.

Fig 4e is less flawed but its interpretation is unpersuasive. Vero cells, I believe, are *Tmprss2* low, which should favor Omicron over Delta (according to the persuasive Fig. 3). So again, Fig. 4e results may again be a simple consequence of lower surface expression of S. (Remember that the live virus buds from the ERGIC not the plasma membrane, so the amount of S protein at the plasma membrane, and incorporated in the retroviral PV, doesn't directly reflect actual virion production). Shouldn't the authors be arguing that syncytia formation limits the burst size of Delta? What are they actually arguing? To quote them, "We predicted that poor fusion would limit cell-cell spread of Omicron". That is, the failure to form syncytia would be disadvantageous for Omicron spread. I might instead interpret these data to suggest that Omicron replicates more efficiently in *TMPRSS2*-low cells because it does not form syncytia that accelerate producer-cell death.

Response: we thank the reviewer for bringing up this area. We agree that the interpretation of focus size is not straightforward, though the data are clear. We have now shown that the Omicron S is well expressed at the surface, and indeed can engage ACE2 efficiently. However, in an assay designed to assess cell-cell infection rather than cell-free, we see smaller foci of infected cells. The cells are in a semi solid media which prevents the cell free infection, hence the focus size reflects the differential cell-cell fusion rather than replication kinetics.

Why is total virion production in the supernatant not measured?

Response: The use of semi solid media precludes this, and we would expect to see no virus budding into the media, only from cell to cell, eg with syncytia. We further show that the smaller foci are due to limited infection at short range using a high magnification automated microscope to measure focus size. We show that at 2hrs post infection the infected focus represents a single cell at around 1 micron. This is similar to the focus size for Omicron after 18 hours, despite opportunity for multiple infection rounds and spread via fusion. For Delta the focus size is three times higher, consistent with cell-cell fusion. These control data with automated microscopy are in the supplementary material.

Summarizing my concerns with Fig, 4: the data point to a model that the delta form syncytia because more functional spike is on the plasma membrane, even if Fig. 4b obfuscates that point. (I could be wrong – but we need the right flow data to know). Even so, we don't have any sense yet whether that is 'good' for Omicron (because syncytia accelerate cell death, limiting burst size) or, as the authors argue, because these syncytia model cell-to-cell transmission, 'good' for Delta. This latter interpretation would suggest that Delta would have higher viral loads in a low-*Tmprss2* setting reflective of the upper respiratory track,

contrary to the population transmission data.

Response: We thank the reviewer for these points and propose to add a section in the discussion where we discuss the findings of smaller foci from the two perspectives: (i) lack of syncytia may increase virus production due to less cell death or (ii) alternatively that syncytia may decrease spread by causing premature cell death. The difference in outcome from syncytia is likely to be cell and tissue dependent. Hence extrapolation of these in vitro data to viral loads in the respiratory tract are likely not straight forward.

In conclusion, the point we would like to make from this figure on focus size is that omicron makes smaller infection foci possibly because of failure to induce syncytia, without compromising on entry efficiency in these cells that are permissive to infection via both endocytosis and plasma membrane fusion. The semi solid media has prevented cell free infection and therefore our measurement should primarily reflect infection spread over a limited time period via syncytia formation. The counting method with high power microscope shows that after 2 hrs the infected focus size is the size of a cell, and after 18 hours Omicrons focus size is around the same, indicating lack of local spread in the semisolid media. In contrast the Delta foci are three fold higher.

Despite these concerns, the paper is important and generally well executed.

Response: we thank the reviewer for the constructive comments, and to note that this work would not be possible without the foundations on SARS biology being established and the early work on SARS-CoV-2 entry and cell-cell fusion. We hope that this work serves as a spring board for others to identify the finer details of molecular mechanisms.

Caught one typo: E64D not ED64 line 283

Response: we thank the reviewer for pointing out the typo

Reviewer 2

The resubmission is significantly improved. Amongst the improvements: (1) superior organization of text (2) removal of incomplete work on MD simulations (3) inclusion of several results documenting differential utilization of TMPRSS2 protease by VOCs.

Given that this is a second round of review, and given that the investigators have responded thoroughly to prior reviews, there are only a few remaining minor comments.

1. Effects of protease inhibitors on live VOC virus infections (Fig 3h): Relative lack of differences between delta and omicron in E64d suppression could arise because E64d blocks the common viral 3CL proteases that operate in viral replication. Suggest indicating limitations to using E64d in that the inhibitor has targets beyond the cathepsins that operate in entry.

Response: we thank the reviewer for this point and will add it to the limitations

2. TMPRSS2 levels and ACE2 levels: A common theme in the manuscript is with omicron failing to use TMPRSS2 for activation (at least not as much as delta), and with ACE levels relating somehow to this failure. For example, see line 399, :TMPRSS2 cofactor usage is impacted by ACE2 levels. One issue is that increasing

TMPRSS2 suppresses ACE2 surface levels. The TMPRSS2 cleaves the ACE2. This appears to be a limitation in the study that might affect data interpretation and might warrant mention in the final paragraph.

Response: we thank the reviewer for this point and will add it to the limitations

3. Patterns of spike protein cleavage: Lines 385-387; proportion of cleaved spike was lower within cells relative to virions. The explanation here is that authentic virion spikes, incorporated at the ERGIC budding site, are cleaved as the virions secrete through exocytic pathways. This is different than the explanation provided.

Response: we thank the reviewer for this point and agree. We have amended the explanation provided to reflect this.